# Facilitating alkaline hydrogen evolution kinetics via interfacial modulation of hydrogen-bond networks by porous amine cages

Shiqi Zhou[1], Wei Cao[2], Lu Shang [3], Yunxuan Zhao [3], Xuyang Xiong[4], Jianke Sun [5], Tierui Zhang [3,6] ✉ & Jiayin Yuan [1] ✉

The electrode-electrolyte interface is pivotal in the electrochemical kinetics. However, modulating the electrochemical interface at the atomic or molecular level is challenging due to the lack of efficient interfacial regulators. Here, we employ a porous amine cage as an interfacial modifier to Pt cluster in a confining configuration, largely enhancing alkaline HER kinetics by facilitating charge transfer. In situ electrochemical surface-enhanced Raman spectra, in combination with the ab initio molecular dynamics simulation, elucidates that the interaction between water and the ·NH· moiety of cage frame softens the H-bonds net of interfacial water, making it more flexible for charge transfer. Moreover, our investigation pinpointed that the ·NH· moiety acted as a pump for charge transfer by Grotthuss mechanism, lowering the kinetic barrier for hydrogen adsorption. Our findings highlight the strategy of establishing a soft-confining interfacial modifier by porous cage, offering opportunities to optimize electrochemical interfaces and promote reaction kinetics in a targeted way.

As a carbon-free energy carrier, hydrogen is considered as an eco-friendly and sustainable fuel to satisfy the ever-growing energy needs of the global economy and to mitigate environmental pollution and climate change[1–3]. Renewable electricity-driven electrolysis of water due to its merits of accessible and abundant resources, carbon neutrality, superior purity and high-output $H_2$ generation, is a potential replacement for fossil fuel-sourced energy technologies[4–6]. The latest advances in inexpensive anion exchange membranes and naturally abundant electrocatalysts for oxygen evolution reaction at the anode enhance the industrial appeal of alkaline water electrolysis[7,8].

Nevertheless, the decrease in kinetics of cathodic hydrogen evolution reaction (HER) by several orders of magnitude at elevated pH largely restricts the advancement of water electrolysis at alkaline conditions[9–13]. Despite decades of research, the atomic and molecular derivation for the pH-dependent kinetics has yet to be uncovered explicitly.

Multiple mechanisms have already been raised to unravel the origin of the pH-dependent HER kinetics. Based on the fact that hydrogen adsorption occurring in the Volmer step is considered as a key step in HER, it is proposed that the pH-dependent H-binding

[1]Department of Chemistry, Stockholm University, Stockholm, Sweden. [2]Frontiers Science Center for Rare Isotopes, Lanzhou University, Lanzhou, PR China. [3]Key Laboratory of Photochemical Conversion and Optoelectronic Materials, Technical Institute of Physics and Chemistry, Chinese Academy of Sciences, Beijing, PR China. [4]Institutes of Physical Science and Information Technology, Anhui University, Hefei, PR China. [5]MOE Key Laboratory of Cluster Science, Beijing Key Laboratory of Photoelectronic/Electrophotonic Conversion Materials, School of Chemistry and Chemical Engineering, Beijing Institute of Technology, Beijing, PR China. [6]Center of Materials Science and Optoelectronics Engineering, University of Chinese Academy of Sciences, Beijing, PR China. ✉ e-mail: tierui@mail.ipc.ac.cn; jiayin.yuan@mmk.su.se

energy is responsible for the pH-dependency of HER kinetics, as reflected by the pH-dependent variation of the cyclic voltammetry (CV) peaks in the potential window of underpotential-deposited hydrogen ($H_{UPD}$) region on the surface of polycrystalline Pt[14–19]. Another widely accepted theory is the bifunctional mechanism which attributes the sluggish HER kinetics to the required high energy barrier to cleave the H-OH bond since the proton source is derived from the dissociation of water ($H_2O + e^- \rightarrow H_{ads} + OH^-$) under alkaline condition, while it is provided by plentiful hydronium ions ($H_3O^+ + e^- \rightarrow H_{ads} + H_2O$) under acidic condition[20,21]. Under the guidance of this proposition, enthusiastic efforts focus on strengthening the OH-binding energy of Pt by mixing high-oxophilic constituents, e.g., hydroxides and Ru to boost the alkaline HER on Pt[22–25].

Despite the important thermodynamic factor in terms of binding strength with reaction intermediates, the electrode process is also unignorably associated with the kinetics of the electrochemical interface, which governs the charge transfer of electrochemical reactions. The Pt(111) single-crystal electrode is a classic model for investigating pH-dependent HER kinetics at electrochemical interfaces. Recent studies have underlined the role of interfacial water structure in pH-dependent HER process from the kinetic aspect[26–31]. It has been found that the strong electric field at high pH rigidifies the interfacial water molecular net, leading to a high energy demand for the reorientation of interfacial water to transport OH- through the electrical double layer[26,27]. Besides the flexibility, the connectivity of the hydrogen-bond (H-bond) net of interfacial water was also reported in relation to the proton transfer and thus, the kinetics of alkaline HER[28]. Therefore, strategies of integrating the inorganic promoters, e.g., Ni, Ni(OH)$_2$[26], and Ru[27,28] with Pt, which could ease the strong interfacial electric field, have been proposed to facilitate the HER kinetics in alkaline conditions by improving flexibility or connectivity of the interfacial water network. Specifically, Pt(111) single-crystal electrodes are often modified with surface promoters like Ni(OH)$_2$ or Ru to steer interfacial water structure. However, these surface promoters also alter or partially occupy the surface sites of Pt. In addition, these inorganic promoters blended with Pt would also change the thermodynamic energetics of Pt, causing the obscured promoting mechanism among thermodynamic or kinetic aspects. An alternative approach involved introducing organic additives into the electrolyte to directly interact with the interfacial water, e.g., caffeine-based[29,30], and N-methylimidazoles[31] to the electrolyte, which could interact with interfacial water. Unfortunately, these organic additives often introduce steric hindrance, which impedes mass transfer through the Pt surface. Moreover, the proximity to the Pt surface of these organic additives implanted into the electrolyte is undeniably challenging to adjust, bringing the dilemma that either the mass transfer to the Pt surface sites was hampered by the inevitably steric effect or the precise interaction with interfacial water is insufficient.

Drawing inspiration from these studies and recognizing the challenges, we intend to employ a non-adsorbed organic modifier in the vicinity of the Pt surface, by which the interference of surface adsorption or steric effect by the modifier could be ruled out so as to enable targeted investigation of the role of the interfacial water in HER kinetics. Herein, we developed an organic porous modifier to modify the Pt surface in a confining configuration in this work. This configuration fully exposes the Pt surface sites while minimizing the steric effect due to the open windows of the 3D porous cage. Additionally, the proximity of the cage modifier to the Pt surface ensures precise interactions with the interfacial water. Furthermore, the confinement effect by cages also induces the formation of ultrafine Pt clusters, which increase the number of exposed Pt sites, thereby enhancing the activity in electrocatalytic reactions and improving atomic utilization of the precious Pt metal[32–34]. By applying this cage-confined Pt catalyst model to HER, we succeeded in boosting the kinetics of the rate-determining Volmer step at a high pH value (Tafel slope of 37 mV dec$^{-1}$

at pH = 13.0). Then, we applied in situ electrochemical surface-enhanced Raman spectra (SERS), in combination with ab initio molecular dynamics (AIMD) simulation to pinpoint cage's function in accelerating the alkaline HER kinetics. The study here unveils that the interplay between the cage and interfacial water primarily occurred through cage's -NH- unit, which lowered the rigidity of the H-bond net of interfacial water at negative HER potentials, and makes it sufficiently soft for convenient rearrangement and better charge transfer. To stress, the -NH- unit functioned as a H$^+$ pump and transferred the formed OH- by building up and cutting off H-bonds with interfacial water. As such, it regenerates the reactive water layer constantly on the Pt surface.

## Results

### Synthesis and characterizations of catalysts

The porous molecular cage-confined Pt clusters of narrowly distributed sizes were synthesized by the electrostatically induced injection and the subsequent in situ reduction strategy (Fig. 1a). Specifically, the ionic cage due to protonated amine units was synthesized in a procedure reported in our previous work[35,36]. Each ionic cage possesses 12 -NH$_2^+$- positive moieties, repulsively dispersing individual cages in an aqueous solution and attracting negatively charged metal precursors through electrostatic interaction[32]. Then, the cage-confined Pt clusters were obtained by in situ reduction by NaBH$_4$, followed by a deprotonation step to restore the 12 neutral -NH- units on the cage molecule. The successful encapsulation of the metal cluster into cage has been well demonstrated in our previous work[35–37].

The morphology of the as-prepared porous cage-confined Pt cluster (termed as "Pt/cage") was first visualized by aberration-corrected high-angle annular dark-field scanning transmission electron microscopy (HAADF-STEM). The Pt existed as ultrafine clusters in an average size of ~0.50 nm in a narrow size distribution (Fig. 1b, and Supplementary Figs. 1 and 2), highlighting the physical confinement effect by the cage frame, in drastic contrast to the control Pt sample synthesized similarly except without adding the cage (Supplementary Fig. 3). It proves the cage-confined Pt clusters are successfully constructed[35]. Powder X-ray diffraction (XRD) pattern was applied to characterize the phase structure of the as-prepared Pt/cage. A broad band located in the range of 20° to 30° existed in Pt/cage, similar to that of the pure cage, suggesting the cage structure is well-maintained (Fig. 1c). In agreement with the HAADF-STEM analysis, the XRD pattern did not observe crystalline Pt phase, as expected for ultrafine Pt clusters below 1 nm.

Synchrotron radiation X-ray absorption spectroscopy (XAS) measurements were performed at the Pt L$_3$-edge to further characterize the electronic structure and coordination environment of Pt/cage. The X-ray absorption near-edge structure (XANES) region of the XAS spectrum carries information on the valence state of Pt. The similar peak positions of white line of Pt/cage and Pt foil revealed the cage-confined Pt clusters existed as Pt$^0$ (Fig. 1d)[38–40]. Further structural coordination information was extracted from the extended X-ray absorption fine structure (EXAFS) spectroscopy. Fourier transformed R-space curves of the Pt L$_3$-edge EXAFS spectra clearly displayed the bonding environment of Pt atoms in Pt/cage. Only an intense peak at 2.8 Å in the Pt/cage was observed (Fig. 1e and Supplementary Table 1), which refers to the Pt-Pt feature when compared with the Pt foil, indicative of Pt-Pt exclusive coordination in Pt clusters of Pt/cage and no other surface-coordination brought by the cage structure[41,42]. Wavelet transform analysis of EXAFS spectra, which could reveal the dependence of k space and R space and thus analyze the back-scattering atoms, was also conducted. The wavelet transformed EXAFS spectra of Pt foil and PtO$_2$ samples present the contour intensity maximal at 11.0 Å$^{-1}$ and 7.5 Å$^{-1}$, representing Pt-Pt and Pt-O-Pt scattering paths (Fig. 1f and Supplementary Table 2), respectively. In the Pt/cage, there is only the same contour intensity with Pt foil, positioned at

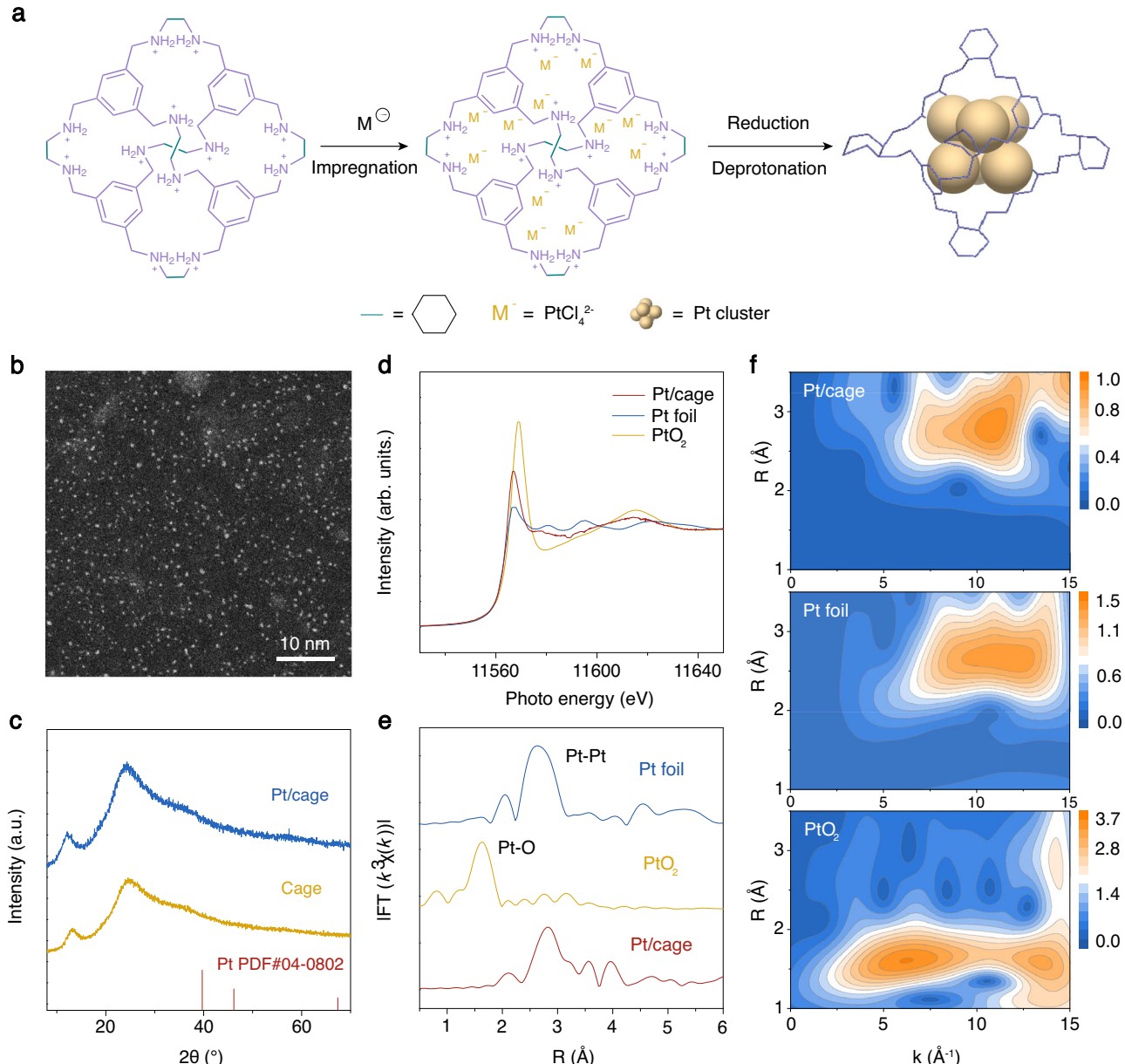

**Fig. 1 | Synthesis and characterizations of catalysts. a** Synthetic procedure of Pt/cage. The counter anion Cl⁻ for the pronated cage is not shown for clarity. **b** HAADF-STEM image of Pt/cage. **c** X-ray diffraction pattern of Pt/cage (blue) and cage (yellow). **d** XANES spectra at Pt L₃-edge for Pt/cage (red), Pt foil (blue), and PtO₂ (yellow). **e** The $k^3$-weighted Fourier transforms of Pt L₃-edge EXAFS spectra for Pt/cage, Pt foil, and PtO₂. $R$ value describes the radial distribution of neighbouring atoms in the shells to the absorbing atom. **f** Wavelet transformed Pt L₃-edge EXAFS spectra for Pt/cage, Pt foil, and PtO₂. $k$ value represents the wave number.

11.0 Å⁻¹, corresponding to the Pt-Pt scattering path, which further supports that no adsorbed species on the surface of the cage-confined Pt clusters, i.e., the Pt cluster being physically trapped in the cage molecules (Supplementary Fig. 4). Moreover, the valence state of Pt atoms was also analyzed by X-ray photoelectron spectroscopy (XPS). The Pt $4f_{7/2}$ and Pt $4f_{5/2}$ of Pt/cage confirmed the same binding energy as the Pt⁰ feature in commercial carbon-supported Pt nanoparticles (termed as 'Pt/C') (Supplementary Fig. 5), further supporting the valence state of 0 for Pt clusters in Pt/cage, in good concordance with the XAS results[43]. Besides, The Pt content of the as-prepared Pt/cage was confirmed by inductively coupled plasma optical emission spectrometry (ICP-OES) to be 5.3 wt.% (Supplementary Table 3). Therefore, the measured XAS results qualify Pt/cage as a reliable model system to probe the interfacial kinetics of HER targeted without any interference

from surface adsorbed species, which may adversely bring energetic changes to Pt in HER.

## Investigation of HER performance and reaction kinetics

The electrocatalytic HER performance of the Pt/cage was evaluated. The commercial Pt/C (20 wt.%) was selected as the control sample (Supplementary Figs. 3 and 6). Performance was investigated by HER polarization curves at a scan rate of 5 mV/s in Ar-saturated alkaline (0.1 M KOH and 0.1 M HClO₄) electrolyte. All the potentials were reported against a reversible hydrogen electrode (RHE). In the polarization curves, Pt/cage exhibited a notable HER activity with much lower overpotential of 32 mV at a current density of 10 mA cm⁻² under alkaline conditions (Fig. 2a, b and Supplementary Figs. 7 and 8), when compared with the commercial 20% Pt/C (overpotential of 64 mV at

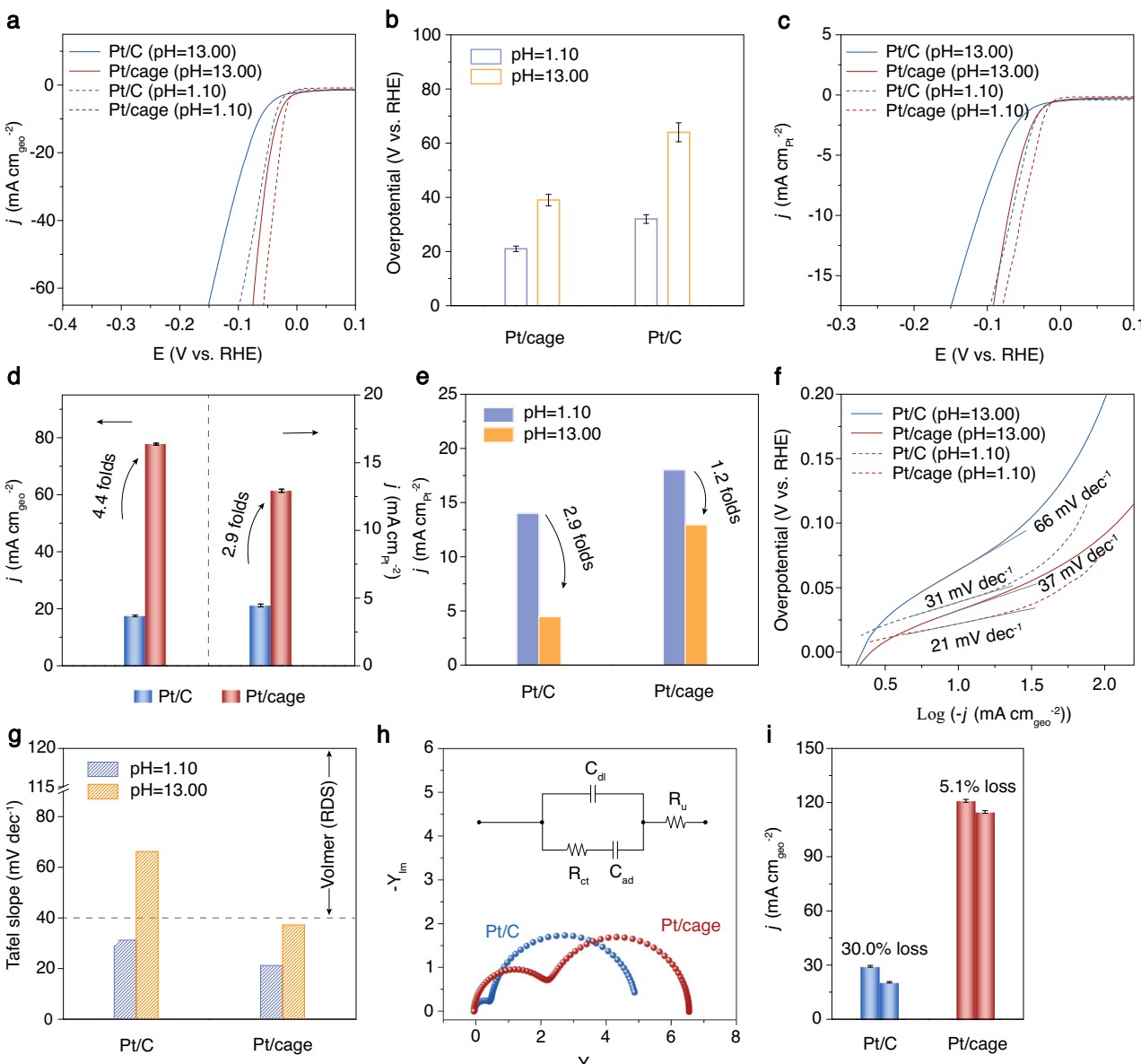

**Fig. 2 | Electrochemical characterizations. a** Polarization curves of Pt/cage (dashed line refers to acidic condition, and red, solid line for alkaline condition) and Pt/C in 0.1 M KOH (pH = 13.00 ± 0.02, the uncompensated solution resistance $R_u$ = 2.31 ± 0.02 Ω) and 0.1 M HClO₄ (pH = 1.10 ± 0.01, $R_u$ = 0.24 ± 0.01 Ω) at 25 °C. The geometric area for the glassy carbon working electrode is 0.196 cm². The Pt loading for Pt/cage and Pt/C working electrodes are 5.4 µg cm⁻² and 5.1 µg cm⁻², respectively. RHE refers to reversible hydrogen electrode. **b** Overpotential values with error bars at 10 mA cm⁻² of Pt/cage and Pt/C samples under acidic (purple) and alkaline (orange) conditions with three independent measurements. **c** ECSA-normalized polarization curves of Pt/cage and Pt/C in 0.1 M KOH and 0.1 M HClO₄. **d** Comparison of current densities normalized by electrode geometrical area and ECSA of Pt/cage (red) and Pt/C (blue) at −80 mV in 0.1 M KOH. **e** Comparison of current densities normalized by ECSA of Pt/cage and Pt/C at −80 mV in 0.1 M KOH

(orange) and 0.1 M HClO₄ (purple). **f** Tafel plots of Pt/cage and Pt/C in 0.1 M KOH and 0.1 M HClO₄. **g** Comparison of Tafel slopes of Pt/cage and Pt/C in 0.1 M KOH and 0.1 M HClO₄. RDS refers to rate-determining step. **h** Admittance Nyquist plots for Pt/cage (red) and Pt/C (blue) in 0.1 M KOH with frequencies from 10 kHz to 0.1 Hz and an amplitude of 5 mV, fitted with the inset equivalent electric circuit for hydrogen adsorption. $C_{dl}$, $R_u$, $R_{ct}$, and $C_{ad}$ refer to the capacitance of the double layer, the uncompensated solution resistance, the charge transfer resistance, and the capacitance from the hydrogen adsorption, respectively. **i** Comparison of current density losses at 100 mV of Pt/cage and Pt/C before and after CV 10 k cycles in 0.1 M KOH (left and right column refers to current density before and after CV cycles, respectively). Error bars represent standard deviation for each data point with three independent experiments.

10 mA cm⁻²). This comparison demonstrates the improved HER activities of Pt/cage in alkaline electrolytes. In consideration of higher utilization of Pt in Pt/cage due to its ultrafine size than Pt nanoparticles in commercial 20% Pt/C, to examine the role of cage structure in HER activity explicitly, the specific activities normalized by electrochemical surface area (ECSA) were compared quantitatively as well (Fig. 2c, d, and Supplementary Fig. 9, Table 4) to exclude the size effect. Pt/cage presented an electrode geometrical area-normalized current density

of 77.72 mA cm$_{geo}$⁻² at −80 mV (vs. RHE), which was 4.4 times to that of Pt/C (17.5 mA cm$_{geo}$⁻²). After excluding the size effect, Pt/cage possessed a notable specific activity of 12.93 mA cm$_{Pt}$⁻², which was 2.9 times to that of Pt/C, implying the enhanced intrinsic alkaline HER activity contributed mainly by the promoting role of the cage, rather than the reduced size of confined Pt clusters. Additionally, the control sample prepared by adding same-amount pure cage into Pt/C catalyst ink showed only slightly improved activity compared with Pt/C

(Supplementary Fig. 10), demonstrating the confining configuration, which ensures the vicinity of cage to Pt surface and thus the interfacial modulation, is critical.

The HER kinetics were then analyzed by comparing the HER performance under acid and alkaline conditions. Owing to the different sources of protons, HER goes through different paths under acidic and alkaline conditions[44]. The elementary steps of HER in acidic aqueous electrolyte are as follows:

$$\text{Volmer step}: H_3O^+ + e^- + {}^* \rightleftharpoons {}^*H + H_2O \qquad (1)$$

$$\text{Heyrovsky step}: {}^*H + H_3O^+ + e^- \rightleftharpoons H_2 + H_2O + {}^* \qquad (2)$$

$$\text{Tafel step}: 2{}^*H \rightleftharpoons H_2 + 2{}^* \qquad (3)$$

While in alkaline aqueous electrolyte, the elementary steps are as follows:

$$\text{Volmer step}: H_2O + e^- + {}^* \rightleftharpoons {}^*H + OH^- \qquad (4)$$

$$\text{Heyrovsky or Tafel step}: {}^*H + H_2O + e^- \rightleftharpoons H_2 + OH^- + {}^* \qquad (5)$$

($^*$indicates the adsorbing site on the catalyst surface)

In acidic electrolyte, protons derive from the enriched hydronium ions. Therefore, the hydrogen adsorption Volmer step is fast and easy to be pre-equilibrated. The non-electron-transferring Tafel step or the second electron-transferring Heyrovsky step becomes the rate-determining step (RDS)[26]. By contrast, in alkaline electrolyte, the adsorbed hydrogen is provided by the cleavage of interfacial water molecules which demand a larger energy for H-OH dissociation and proton transfer, resulting in the hindered kinetics of hydrogen adsorption reaction. In this case, the Volmer step is the RDS. Pt/C showed a substantial drop of specific activity from acidic condition to alkaline condition (Fig. 2e) due to insufficient protons supply, while Pt/cage only presented a slight decrease in specific activity. The specific activity of Pt/cage under alkaline condition was comparable to that of Pt/C under acidic condition, exhibiting an acid-like HER performance that boosted by cage. This revealed that cage largely alleviated the decrease of HER kinetics in alkaline media presumably by promoting the charge transfer at the Pt-electrolyte interface.

The HER kinetics were further elucidated by the Tafel plots derived from the polarization curves, by which the information about the rate-determining step is analyzed. The Pt/C showed a large Tafel slope of 66 mV dec$^{-1}$ at 0.1 M KOH electrolyte (Fig. 2f, g), demonstrating the Volmer-limited HER kinetics[45]. For Pt/cage, the Tafel slope value was largely lowered to 37 mV dec$^{-1}$ in 0.1 M KOH, which is comparable to that of Pt/C in acid, indicating an acid-like RDS and largely promoted hydrogen adsorption reaction with the manipulation by cage structure. Considering there is no interference variable coordinating with Pt clusters observed that may adversely change the surface adsorption strength of Pt by the results from Fig. 1d, e, and the major contribution to the enhanced alkaline HER activity from the cage structure, the fast Volmer step kinetics under alkaline condition on Pt/cage was accordingly attributed to the assistance of the cage structure, which is involved the charge transfer process. Then, the charge transfer process was investigated by electrochemical impedance spectroscopy (EIS). From the admittance Nyquist plots fitted with the equivalent circuit of hydrogen adsorption reaction (Fig. 2h), $C_{dl}$, $R_u$, $R_{ct}$, and $C_{ad}$ refer to the capacitance of the double layer, the uncompensated solution resistance, the charge transfer resistance, and the capacitance from the hydrogen adsorption, respectively. A much larger radius of the first semicircle of the admittance Nyquist plot was observed for Pt/cage, corresponding to a much lower charge transfer resistance, than that of Pt/C, pointing out the cage structure located at

the interface between Pt surface and interfacial water largely facilitated the charge transfer during the process of hydrogen adsorption under alkaline condition. Additionally, the performance of Pt/cage in this work has been compared with previous similar studies. Pt/cage exhibited the alkaline HER activity that lies among the reported top-level noble-metal electrocatalysts drop-casted on flat glassy carbon electrodes (Supplementary Table 5)[46–52].

The chemical stability of Pt/cage was assessed by comparison of the current density at 100 mV before and after 10k cycles of tests (Fig. 2i and Supplementary Fig. 11). Pt/cage exhibited a much higher current density of 127 mA cm$^{-2}$ in 0.1 M KOH, than that of Pt/C (28 mA cm$^{-2}$). Furthermore, the current density loss of Pt/cage was only 5.1% after 10k cycles, much lower than that of Pt/C. In addition, no change of valence state and size was observed by XPS and HAADF-STEM characterizations after the cycle tests; so is the same as the phase structure of cage-confined Pt clusters as verified by XRD tests (Supplementary Figs. 12, 13, and 14). It embodies the structural robustness of Pt/cage as a result of individual cage's physical confinement of Pt clusters.

## Probing interfacial water structure by in situ electrochemical SERS

Interfacial water plays an important role in HER kinetics. During the alkaline HER process, the protons to be adsorbed on the Pt surface are provided from the bulk water molecules to the interfacial water layers via transfer through the H-bond net of water, for which the reorganization of the dynamic H-bonded water net and water molecules' dynamic rotation are the prerequisite[26,27,53–55]. Thus, the flexibility of the H-bonded net of water molecules in the HER kinetics plays a crucial role. Within the potential window of HER, the rather negative electric field, which is generated at the interface between electrode and electrolyte, endows the interfacial water net with higher order and rigidity, for which a higher reorientation energy is required for the interfacial water in charge transferring, naturally retarding the HER kinetics. Notwithstanding the pivotal role of interfacial water structure in HER kinetics, the strategy to solely play with the interfacial water structure by constructing a proper catalyst model system and the corresponding techniques to probe the dynamic evolution of interfacial water structure is still under active investigation.

Different from infrared spectroscopy, Raman spectroscopy records weak Raman signals for water. That makes Raman spectroscopy more feasible to detect interfacial water structure by excluding the interference of bulk water when combined with surface-enhanced techniques[56]. Surface-enhanced Raman spectroscopy (SERS) in combination with the shell-isolated nanoparticles (SHINs) that could yield a strong coupling effect is a capable method to approach detailed structure information of only interfacial water in the proximity of electrode surface within the scope of enhancement[57,58]. For this consideration, we conducted in situ electrochemical SERS measurements with pinhole-free Au@SiO₂ SHINs (Fig. 3a, and Supplementary Figs. 15 and 16) to investigate the cage-interfacial water interactions in the alkaline HER process. Since the stretching vibration of water could unveil more structural information due to its sensitivity to environmental changes and the intensity of the stretching vibration signal is much higher than that of bending vibration or librations, we focused on the stretching vibrational modes of interfacial water. The background Raman spectrum of the sample was subtracted. The obtained Raman spectra show the vibrational Stark effect at the applied potentials, implying the Raman signals are mainly contributed by the interfacial water in the immediate proximity of the electrode surface (Fig. 3b, c). The broad stretching vibrational band is positioned from 3000 cm$^{-1}$ to 3800 cm$^{-1}$ in the Raman spectra. Gaussian fitting was applied to deconvolute it into three O-H stretching modes of interfacial water, i.e., -3600 cm$^{-1}$ for stretching of free water located, -3400 cm$^{-1}$ for asymmetric stretching vibration of weak H-bonded

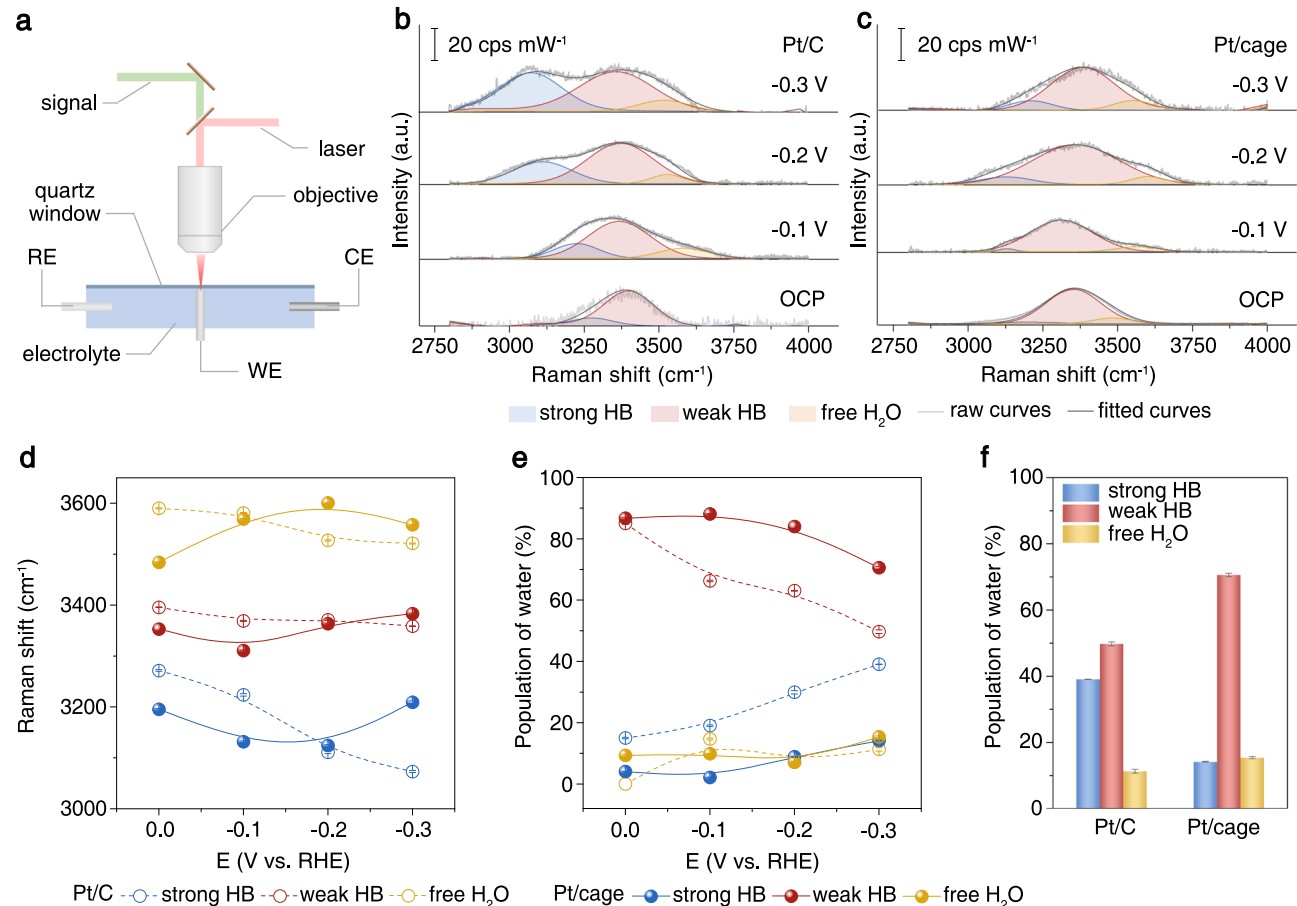

**Fig. 3 | In situ electrochemical SERS characterizations. a** Schematic illustration of the in situ electrochemical SERS setup (WE refers to the working electrode, RE refers to the reference electrode, CE refers to the counter electrode). **b** In situ electrochemical SERS spectra of the O·H stretching vibrational mode of Pt/C in 0.1 M KOH (HB refers to H-bonds, OCP refers to open circuit potential, cps refers to counts per second). **c** In situ electrochemical SERS spectra of the O·H stretching vibrational mode of Pt/cage in 0.1 M KOH. **d** Potential-dependence of the Raman shifts of the three deconvoluted peaks of interfacial water on Pt/C and Pt/cage, respectively. **e** Potential-dependence of the ratio of the three deconvoluted peaks of interfacial water on Pt/C and Pt/cage, respectively. **f** Comparison of the ratios of the three deconvoluted peaks of interfacial water on Pt/C and Pt/cage at an applied potential of −0.3 V. Strong HB (blue), weak HB (red), and free water (yellow). Error bars in (**d**, **e**, **f**) represent standard deviation for each data point with three independent experiments.

water, and ~3200 cm⁻¹ for symmetric stretching vibration of strong H-bonded water. Free water molecules are more accessible to rotation, but the poor connectivity of the network is inefficient for proton transfer, while a too strongly H-bonded water network with good connectivity is rigid and demands a high energy for reorientation to transfer protons. When the evolution of frequencies and populations of the three different H-bonded water systems are compared (Fig. 3d, e), all three stretching vibrations have obvious redshifts, especially the large shift shown by the stretching vibration of strong H-bonded water from 3272 cm⁻¹ to 3074 cm⁻¹. Meanwhile, the population of strong H-bonded water has largely increased, as observed in the Raman spectra of Pt/C upon the increase of applied negative potential. It indicates the H-bond net of interfacial water on the Pt/C surface becomes more ordered and rigid due to the increased negative electric field, which implies a higher energy barrier for the reorientation of interfacial water for charge transfer on Pt/C. However, the frequencies show only slight shifts and the weak H-bonded interfacial water still dominated even at an applied negative potential of −0.3 V in the presence of cage structure on Pt/cage (Fig. 3f), suggesting the cage intervenes in the H-bonded net of interfacial water, presumably by its -NH- moiety, which readily forms moderate H-bonds with interfacial water molecules, and thus holds a moderate H-bonded structure which is easier to reorganize for charge transfer.

## Investigation of the role of cage by AIMD simulations

We further conducted AIMD simulations to investigate the role of cage in alkaline HER. The cage-confined Pt cluster-water interface was built to simulate the Pt/cage (Supplementary Data 1). The amorphous Pt cluster was constructed by six Pt atoms according to their average size and in a close-packing arrangement tentatively, in which the positions of the Pt atom were optimized as randomly arranged during AIMD. Pt(100)-water interface was selected as the model to simulate the polycrystalline Pt nanoparticles in Pt/C, rather than the saturated-coordinated Pt(111), which has a unique hydrogen and hydroxide adsorption behavior. The alkaline condition was simulated by introducing net electron charges to the Pt surface. The configuration and distribution of water molecules on Pt surface were examined by the z-axis distribution on Pt(100) and the radial distribution on Pt/cage to reduce the difference brought by the models as much as possible (Supplementary Figs. 17 and 18). The first peak of Pt-H is located at a shorter distance than that of Pt-O, indicating the H side of the water molecule points to the Pt surface due to the negative electric field at the Pt-water interface (Fig. 4a, b). Furthermore, the similar position of the first Pt-O peaks from the distribution function of Pt(100) and Pt/cage refers to a similar distance between the Pt surface and the first-layer water molecules at Pt(100)-water interface and Pt/cage-water interface, implying that there is no steric effect caused by the

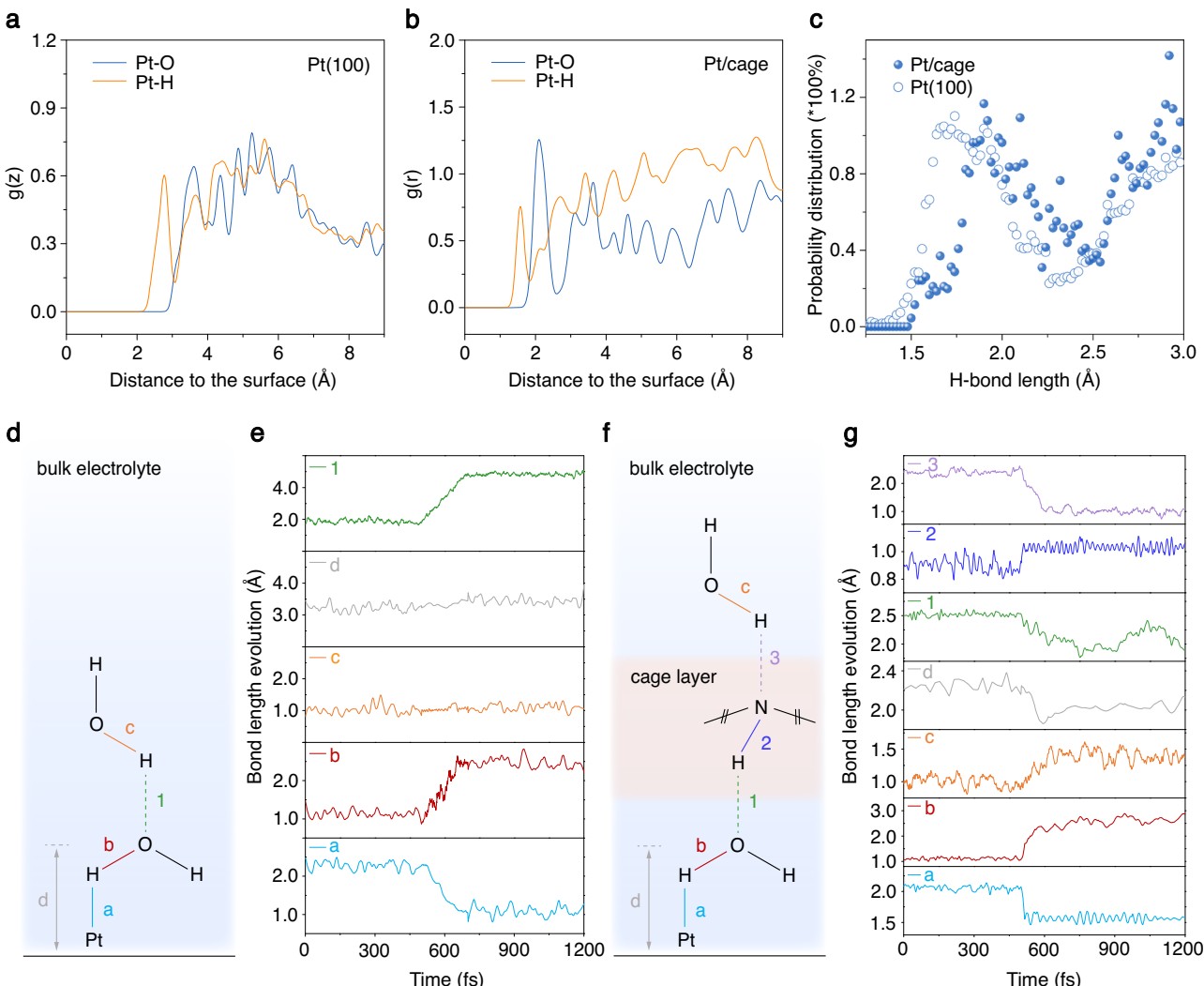

**Fig. 4 | AIMD simulations. a** Distribution function of O (blue) and H (yellow) on the surface of Pt(100) along the surface normal direction z. **b** Distribution function of O and H on the surface of cage-confined Pt cluster along the surface radial direction r. **c** Probability distribution of H-bonds with different bond lengths on Pt(100) (hollow symbol) and Pt/cage (solid symbol). **d, f** Schematic illustration of the interaction between interfacial water molecules and Pt surface atoms at Pt(100)-water interface and Pt/cage-water interface during the alkaline HER process, respectively. N·H simply represents the -NH- moiety of cage. The left part of the cage frame is not presented here for clarity. **e, g** The bond distances evolution along with the time during the alkaline Volmer reaction at Pt(100)-water interface and Pt/cage-water

interface, respectively. The bonds correspond to the bonds in the schematic illustration in (**d, f**) with the same labels, respectively. For Pt(100)-water interface model, Pt-H bond (light blue), Pt-O bond (grey), O-H bond in the first-layer water molecule (red), H bond between the water molecules in the first layer and the second layer (green), and O-H bond in the second-layer water molecule (orange). For Pt/cage-water interface model, Pt-H bond (light blue), Pt-O bond (grey), O-H bond in the first-layer water molecule (red), H bond between the water molecules in the first layer and -NH- moiety in cage (green), N-H bond in the -NH- moiety (deep blue), H bond between the -NH- moiety and the second-layer water molecules (purple), and O-H bond in the second-layer water molecule (orange).

molecular cage frame at Pt/cage-water interface owing to the open-window structure of cage. Moreover, the relative position of -NH-moiety on the interfacial water region is suggested as the layer between the first-layer water and the second-layer water by the Pt-N distribution function (Supplementary Fig. 19). Next, the probability distribution function of H-bonds at different lengths with and without cage was compared to get insight into the interaction between the -NH-moiety and the surrounding H-bonded net of water molecules theoretically (Fig. 4c). The H-bonds at Pt/cage-water interface distributed at longer lengths than that of Pt(100)-water interface, suggesting the -NH- moiety gets involved in the H-bond network between the first- and the second-layer water on Pt surface, and weakened the strength of the H-bonds net, which is in consistent with our in situ electrochemical SERS experimental results, probably owing to the lower electronegativity of N atom than O atom when acting as the H-bond

donor and acceptor (Supplementary Fig. 20). A relative weak H-bonded net of water caused by the -NH- moiety could minify the rigidity increase under the negative electric field owing to the reduced dipole between the H-bond donor/acceptor and H atom, which consequently is more flexible for reorientation for the charge transfer during alkaline HER process and consistent with the experimental observations that the Volmer step is facilitated on Pt/cage.

To unveil how -NH- moiety gets involved in the charge transfer during the alkaline HER process, the evolution of bond distances along with the Volmer reaction was analyzed according to the relative position of -NH- and interfacial water molecules obtained from the distribution function (Fig. 4d, f)[31,59,60]. As shown, after a negative potential is applied, there is a sharp reduction in the bond distance of Pt-H to ~1 Å along with the extension of the H-O bond distance to ~2.5 Å (Fig. 4e, g, and Supplementary Fig. 21), suggesting the formation of Pt-H bond and

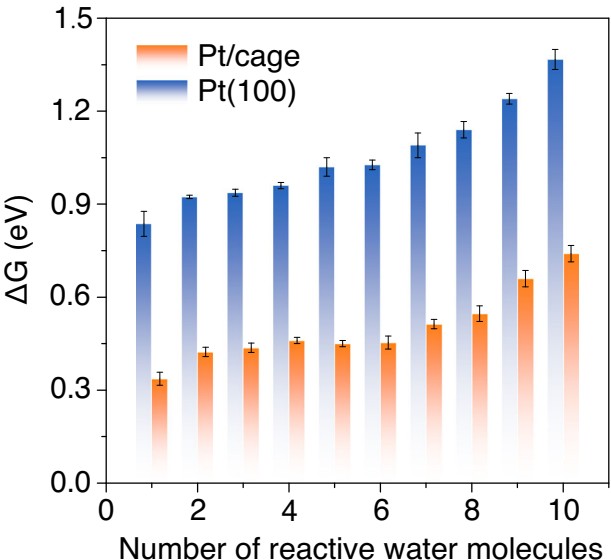

**Fig. 5 | Plot of kinetic barriers of the Volmer step.** The comparison of kinetic barriers ($\Delta G$) of individual Volmer reaction on Pt(100) (blue) and Pt/cage (orange) with different numbers of closest reactive water molecules. Error bars are calculated with three individual parallel meta-dynamics simulations.

the breaking of H-O bond in the reacted water molecule in the first layer, which corresponds to the formation of adsorbed hydrogen in the Volmer step of alkaline HER. After the Volmer step, there is no apparent interaction between the generated hydroxide and the water molecule in the second layer on Pt(100), implying an inefficient charge transfer by the rigid H-bond net. However, in the case of Pt/cage, the interactions between the -NH- moiety, the generated hydroxide, and the water molecule in the second layer are observed after the Volmer step. Specifically speaking, the -NH- tends to transfer its H to the generated hydroxide through the H-bond as a donor to form a sub-stitute reactive water on Pt surface for subsequent reaction, and meanwhile obtain another H from the water molecule in the second layer as an H-bond acceptor, as the bond distances of 1, 3 decreased and bond distances of 2, c increased, respectively. During this process, the -NH- acts as a pump that continuously injects protons onto Pt surface and simultaneously obtains protons through the H-bonds net from the second-layer water so as to release generated hydroxide out by the Grotthuss mechanism[61,62], that is, by H-bonds forming and breaking. The smooth pumping process is ascribed to the assistance of -NH- moiety in the vicinal cage frame, rewiring the interfacial H-bonds net of the proper flexibility, making it easier for reorganization to accommodate and transfer the protons and the generated hydroxides, than the rigid H-bonded water net under applied negative potentials on Pt(100), thus accelerating the HER kinetics. The comparison of the calculated kinetic barriers for individual hydrogen adsorption reactions further verified the promoted kinetics of rate-determining Volmer step on Pt/cage (Fig. 5). Pt/cage exhibited a much lower barrier of 0.34 eV for the Volmer step than 0.84 eV of Pt(100), even with more reactive water molecules, suggesting the facilitated HER kinetics by the assistance of cage located at the Pt-electrolyte interface which well-explained the promoted HER kinetics in the experiments.

Climbing-image nudged elastic band calculations have been performed to analyze the free energy changes for the whole HER. The solvation effect was also considered with VASP sol++ for the calculation to include the interactions derived from electrostatics, cavitation, and dispersion. Following the Volmer step, the Heyrovsky step was chosen as the second elementary step according to the Tafel slope

values in our experimental outcome. It revealed that the HER becomes Heyrovsky-limited on Pt/cage under alkaline conditions. An energy barrier of 1.62 eV to overcome for the Volmer step on Pt(100) implies the Volmer step is the rate-determining step of the whole HER on Pt(100) (Supplementary Fig. 22). However, this barrier is largely decreased and lower than that of Heyrovsky step (1.25 eV) on Pt/cage model, suggesting the easier Volmer step followed by the rate-determining Heyrovsky step on Pt/cage. The findings were in good consistency with our experimental outcome, which indicates an acid-like RDS and largely promoted hydrogen adsorption reaction with the manipulation by cage structure. Moreover, the theoretical over-potentials for HER on Pt/cage and Pt(100) were also calculated and analyzed (Supplementary Fig. 23). Pt/cage presented a lower theoretical overpotential of 0.125 V than that of Pt(100) (0.295 V), which agrees well with our experimental observations.

## Discussion
To conclude, we discover a porous amine-functionalized organic cage serving as a molecular modifier of Pt in a confined configuration to modulate the electrochemical interface of alkaline HER. The alkaline HER kinetics is successfully facilitated by this porous cage-confined Pt clusters catalyst. By combining electrochemical investigation, in situ electrochemical SERS, and ab initio molecular dynamics simulation, we acquire a comprehensive understanding of the facilitating mechanism of alkaline HER kinetics by using this porous cage, which interposes the H-bond net of interfacial water by forming moderate H-bonds between its -NH- moiety and interfacial water molecules, thus allowing for more flexible reorganization of water during charge transfer. Furthermore, the -NH- moiety of the cage frame performs as a pump to transfer protons in continuously and the generated hydroxides out through the electrical double layer with this flexible H-bond net via the Grotthuss mechanism, refreshing the reactive water layer on the Pt surface. Our results underscore the importance of the interface during electrochemical reactions, and the strategy of applying the interfacial modifier with a soft-confining configuration presented in this work offers a paradigm for guiding the design and manipulation of more electrochemical interfaces.

## Methods
### Chemicals and materials
Sodium borohydride (NaBH$_4$, Acros Organics, 99%), potassium tetrachloroplatinate (II) (K$_2$PtCl$_4$, Alfa Aesar, Pt metals basis 99.99%), graphite nanoplatelets (Thermo Scientific Chemicals), potassium hydroxide (KOH, Sigma-Aldrich, trace metal basis 99.99%), gold chloride trihydrate (HAuCl$_4$·3H$_2$O, Sigma-Aldrich, Au basis ≥49.0%), dichloromethane (CH$_2$Cl$_2$, Sigma-Aldrich, ≥99.9%), perchloric acid (HClO$_4$, 70%), sodium silicate solution (Sigma-Aldrich, 27% wt./wt.), 3-aminopropyl trimethoxysilane (Thermo Scientific, 97%), trisodium citrate dihydrate (VWR, 99.8%), hydrochloric acid (HCl, 37%), hydrogen (Strandmöllen, Sweden, lab line 5.0) and argon (Strandmöllen, Sweden, lab line 5.0) were all used as received and without further purification. Milli-Q water (18.2 MΩ cm, at 25.4 °C) was prepared by Milli-Q IQ 7000.

### Synthesis of Pt/cage
To prepare Pt/cage, I-cage-Cl (15 mg) as a precursor (chemical structure shown in Fig. 1) was dissolved in 9 mL milli-Q water (to give Solution I). K$_2$PtCl$_4$ (1.06 mg) was dissolved in 0.5 mL milli-Q water (to give Solution II). NaBH$_4$ (1.00 mg) was dissolved in 0.5 mL milli-Q water (to give Solution III). KOH (0.56 mg) was dissolved into 1 mL milli-Q water (to give Solution IV). First, Solution II was added into Solution I and the mixture solution was aged for 5 min. Then, Solution III was added into the mixture with vigorously shaking for 2 min. Solution IV was finally dropped into the mixture within 2 min under shaking. After the complete addition of Solution IV, the mixed solution was kept

shaking for 30 min. The as-prepared Pt/cage was obtained by centrifugation (6439 × $g$, 15 min), washed by milli-Q water for three times, and finally freeze-dried.

## Synthesis of 55 nm Au@ 2 nm SiO$_2$ SHINs

Au@ 2 nm SiO$_2$ SHINs was synthesized according to Li et al.'s work[58]. To prepare the Au nanoparticles of 50 nm in size, gold chloride trihydrate (23.2 mg) was dissolved in 200 mL milli-Q water (to give Solution V). trisodium citrate dihydrate (22.8 mg) was dissolved in 2 mL milli-Q water (to give Solution VI). 3-aminopropyl tri-methoxysilane (7.2 µL) was dissolved into 40 mL milli-Q water (to give Solution VII). A hydrochloric acid solution was added into 2 mL sodium silicate solution to adjust its pH value to 10 (pH value was measured by pH meter, 24 °C). Then milli-Q water was added till the volume to 100 mL (to give Solution VIII). 200 mL of Solution V was firstly boiled, to which 1.4 mL of Solution VI was quickly added. The mixture solution was kept boiling with flux for 1 h. After cooling naturally to room temperature of 25 °C, the dispersion of Au nanospheres of 50 nm in size was obtained. Next, 0.4 mL of Solution VII and 3.2 mL of Solution VIII were added into 30 mL of the undiluted Au nanosphere dispersion. After that, the mixture solution was stirred at 99 °C for 30 min to get the SHINs dispersion. The SHINs dispersion was centrifugated at 402 × $g$ for 10 min, and then the supernatant was removed. The residue solid was repeatedly for three times. Finally, the residue was dispersed into 200 µL milli-Q water to get the Au@SiO$_2$ stock dispersion.

## Materials characterizations

The high-angel annular dark-field scanning transmission electron microscopy (HAADF-STEM) photographs were recorded with an aberration-corrected transmission electron microscope (Thermo Fisher Scientific Themis Z). The accelerating voltage under operation was set to 300 kV. The energy-dispersive X-ray spectroscopy (EDS) photographs and the corresponding elemental mapping were collected on a SuperX EDS detector on the Themis Z device. The X-ray diffraction (XRD) patterns were taken under a Cu K$_\alpha$ radiation (40 kV, 40 mA, $\lambda$ = 1.5418 Å) in a 2θ range from 5° to 70°. The content of metal species was confirmed by inductively coupled plasma optical emission spectrometry (Agilent ICPOES730) operated under an Ar environment. X-ray photoelectron spectroscopy surveys (XPS, Thermo Fisher, Escalab 250XI, America) were collected from a monochromatic Al-K$_\alpha$ X-ray source (hυ = 1486.6 eV) at an operational power of 150 W.

## X-ray absorption spectra data analysis

The spectra of the XANES and the EXAFS were collected at the Pt L$_3$-edge. The tests were carried out in a fluorescence mode on the BL14W1 Beamline at Shanghai Synchrotron Radiation Facility (SSRF).

The original data were treated by a standard procedure using the open-source Athena program[63]. The wavelet transform procedure of the EXAFS spectra were conducted with the Hama Fortran program[64,65], We chose Morlet wavelet to conduct the transformation for all samples with the following settings of parameter: $R_{min}$ = 0, $R_{max}$ = 6, $\sigma$ = 1, and $\kappa$ = 6.

## Electrode preparation

The Pt/cage working electrode was fabricated as follows. Initially, Pt/cage (0.5 mg), a Nafion dispersion 1-propanol (30 µL, 5 wt.%), and graphite nanoplatelets (2 mg) were mixed in a 1 mL mixture solution of ethanol/water (v/v ~ 1:1). After sonication for 30 min, a catalyst ink was ready. Next, the Pt/cage working electrode was prepared by dropping a 40 µL portion of the ink onto the glassy carbon working electrode, which after drying ended up with a Pt loading of 5.4 µg cm$^{-2}$. A similar procedure was applied to fabricate the Pt/C working electrode, except 0.5 mg of Pt/C (Pt content of 20 wt.%) was dropped into a 4 mL mixture solution of ethanol/water (v/v ~ 1:1), ending up with a Pt loading capacity of 5.1 µg cm$^{-2}$.

As for in situ electrochemical Raman measurements, a similar procedure was applied to fabricate the working electrodes, except 2 µL of the previously made Au@SiO$_2$ SHINs stock solution was dropped onto the electrodes.

## Electrochemical measurements

A three-electrode cell was used to carry out all electrochemical measurements. The measurement data were recorded on a Bio-Logic SAS electrochemical workstation in the model of VSP−300. The counter electrode is a graphite rod; the reference electrode is a Ag/AgCl electrode saturated by KCl in milli-Q water, which is calibrated by placing it and a Pt wire electrode in an electrolyte of sulfuric acid at 0.5 mol/L under H$_2$ bubbling; the working electrode is a glassy carbon electrode in a diameter of 5 mm. The cathodic and anodic compartments were separated by proton exchange membrane (Nafion 117) and anion exchange membrane (Fumasep, FAA−3PK-130) under acidic and alkaline conditions, respectively. Before each measurement, the cell was deep cleaned and washed with milli-Q water thoroughly. The glassy carbon electrode surface was cleaned by soft polishing with alumina of 0.05 µm, and washed in ethanol and milli-Q water with sonication for 5 min, respectively, to ensure the accuracy of each electrochemical measurement. The electrolyte was prepared with milli-Q water and perchloric acid or potassium hydroxide freshly for experiments. The pH values of freshly prepared electrolyte were obtained by pH meter with three independent measurements. The polarization curves were collected without $i$R correction. By using the following equation, we rescaled the electrode potentials according to the RHE:

$$E_{RHE} = E_{Ag/AgCl} + 0.1976 + 0.0591 \times pH$$

A scan rate of 20 mV s$^{-1}$ in 0.1 M KOH solution (Ar-saturated, pH = 13.00 ± 0.02) at a temperature of 25 °C was applied to collect the CV data. A scan rate of 5 mV s$^{-1}$ in 0.1 M KOH solution (Ar-saturated, pH = 13.00 ± 0.02) and 0.1 M HClO$_4$ solution (pH = 1.10 ± 0.01) was used to record the polarization curves at 25 °C.

EIS measurements were measured with frequencies from 10 kHz to 0.1 Hz with an amplitude of 5 mV, and the data were fitted to the equivalent electric circuit by using ZSimpWin software.

Nafion 117 membrane was pretreated by boiling it in 3% hydrogen peroxide solution for 1 h (80 °C), deionized water for 2 h (80 °C), and finally 0.5 mol/L sulfuric acid solution for 1 h (80 °C). The membrane was rinsed between steps and after the final step, and then stored in deionized water. Anion exchange membrane was pretreated by immersing it in 0.5 mol/L sodium chloride solution for 72 h (25 °C) with refreshing the solution several times (stabilized by two glass meshes to avoid curling), and then transferring it into hydroxide form by immersing it in 0.5 mol/L KOH solution at least for 24 h. The membrane was stored in 0.5 mol/L KOH solution in a sealed container, and was rinsed by deionized water before using.

## Electrochemical surface area (ECSA)

The electrochemical surface area (ECSA) was qualified based on the Coulombic charge of the hydrogen underpotential deposition (H$_{UPD}$) (H$^+$ + e$^-$ + $^*$ → $^*$H) region (0.05−0.45 V) in the CV curves[66–69]. The Ar-flow was continuously supplied to avoid oxygen reduction reaction or interferes derived from dissolved oxygen.

## In situ electrochemical SERS measurements

In situ electrochemical Raman measurements were performed at a resolution scale of 0.5 cm$^{-1}$ on a LabRAM HR 800 Raman spectrometer (Paris, France) with from an air-cooled double-frequency laser an excitation wavelength of 785 nm, and a portable Gamry Interface 1010e electrochemical workstation. The band frequency of 520.7 cm$^{-1}$ of a silicon wafer was set as the frequency of the Raman spectrum by calibration. The SHINs carrying a Au nanosphere core of 55 nm in size

and a $SiO_2$ shell of 2 nm in thickness was casted onto the working electrode to acquire the corresponding surface-enhanced Raman signal (the pinhole tests of as-prepared SHINs are described in detail in Supplementary information). Electrochemical measurements were conducted in a in situ electrochemical Raman cell (Gaoss Union, C031–3, China) with anion exchange membrane (Fumasep, FAA−3PK-130) for cathodic and anodic compartments separation, using Chronoamperometry method with different applied potentials. The Raman spectra were recorded after the current was stabilized for 5 min at each potential.

## AIMD simulations

Density functional theory (DFT) was applied to carry out the theoretical calculations, and was performed via the Vienna ab initio simulation package (VASP)[70,71]. The electron exchange and correlation energy were handled by the generalized gradient approximation in the Perdew-Burke-Ernzerh of functional (GGA-PBE)[72]. The plane-wave basis sets bearing cutoff energies of 400 eV were applied to describe the valence orbitals of H ($1s$), C ($2s$, $2p$), N ($2s$, $2p$), O ($2s$, $2p$) and Pt ($5d$, $6s$), using the Gaussian smearing scheme in a smearing width of 0.05 eV. We take the *van der Waals* interaction into consideration via the semi-empirical D3 dispersion correction scheme of Grimme[73]. All simulations include non-spin polarization, as it has little-to-no impact on the entire energies of simulations. We set k-point sampling for all models by using the Monkhorst-Pack scheme bearing a ($1 \times 1 \times 1$) mesh. $10^{-5}$ eV was used as the convergence criteria for the electronic self-consistent iteration; and $0.05 \, eV \cdot Å^{-1}$ for the force. All AIMD simulations employ 1.0 fs as the time step. To maintain the canonical ensemble at 300 K, a Nose-Hoover thermostat was used.

To modeling the Pt(100)/water interface, 71 water molecules were added above a four-layer $4 \times 4$ orthogonal Pt(100) slab in a surface area of $1.1 \, nm^2$.

The water film is about 1.70 nm in thickness, and the whole simulation box is about 2.59 nm in size in the z-axis. To build up the Pt/cage, 6 Pt atoms in a fluster form were placed in the cage with a molecular formular of $C_{72}N_{12}H_{108}$. The aforementioned object was placed in a box of $24.32 \times 24.32 \times 24.32 \, nm^3$ in size. In total, the space of the system was to host 338 water molecules, so that its density of water is similar to that of the Pt(100)/water model. The water film was controlled to carry a density of $0.97 \, g/cm^3$. Molecular dynamics simulation (reaction force field at 300 K) employing the Large-scale Atomic/Molecular Massively Parallel Simulator (LAMMPS) with user package Reax/C was carried out for 150 ps; this period was necessary to completely equilibrate the H-bond networks of water film and to reduce the time for AIMD simulation[74,75]. The reactive force field for the case of Pt-O-H raised by Shin et al. was employed to depict the water-Pt surface interaction[76]. The pre-equilibrated models were first obtained from classical molecular dynamic simulations; they were in turn utilized as the initial configurations for the AIMD simulations. All AIMD trajectories are sampled for up to 20 ps ofruntime so as to secure the fully equilibrated state of the systems. For statistical analyses, the last 10 ps snapshots were implied.

The next effort is to approach the kinetic barrier of the Volmer step. we conducted a meta-dynamics simulation and used the last atomic configurations and velocities in the equilibration. We defined collective variable (CV) as the difference between two atomic distances:

$$CV = d_{Pt-H} - d_{O-H}$$

Here $d_{Pt-H}$ stands for the distance between the H atom of the interfacial water and the surface Pt atom and; $d_{O-H}$ represents the water bond length.

A time-dependent bias potential with Gaussian height ($h$) of 0.08 eV and width ($w$) of 0.18 eV was introduced at an interval of 20 fs.

A single Gaussian hill bearing $h = 1.0$ eV and $w = 0.2$ eV was used to modulate the meta-dynamics so it does not go beyond a CV of 2 angstroms. It prevents the diffusion of water from the interface to bulk without Volmer reaction. By summing the Gaussian potential after the barrier crossing, we calculate the kinetic barrier.

## Data availability

All data that supporting this study are provided in the main text and the Supplementary information. Source data are provided with this paper.

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

## Acknowledgements

L.S., Y.Z., and T.Z. thank the financial support from the National Key Projects for Fundamental Research and Development of China (2023YFA1507201). S.Z. and J.Y. are grateful for financial support from the Stockholm University Strategic Fund SU FV-2.1.1-005, the Wallenberg Academy Fellow Prolongation program (Grant KAW 2022.0194) from the Knut & Alice Wallenberg Foundation in Sweden. L.S., Y.Z., X.X., and T.Z. thank the financial support from the National Key R&D Program of China (2021YFA1500803), the National Natural Science Foundation of China (52120105002, 52202198), and the CAS Project for Young Scientists in Basic Research (YSBR-004). We thank Shihui Feng, Dr. Cheuk-Wai Tai, and Dr. Anumol Ashok at Stockholm University for the HAADF-STEM characterizations. We thank Dr. Shuo Chen for X-ray absorption fine structure spectra measurements on the BL14W1 Beamline in Shanghai Synchrotron Radiation Facility (SSRF).

## Author contributions

S.Z. conceived the idea for the project and designed the experiments. S.Z., W.C., and J.S. conducted the synthesis. S.Z., X.X., and Y.Z. performed the structural characterizations. S.Z. conducted the electrochemical measurements and in situ electrochemical surface-enhanced Raman spectroscopy measurements. S.Z. and L.S. proposed the electrocatalytic mechanism. S.Z. planned the computational simulation, processed the data analysis, prepared the Figures, and wrote the original manuscript. All authors contributed to the revision of the manuscripts. T.Z. and J.Y. supervised the project.

## Funding

## Competing interests

The authors declare no competing interests.
