## [Transparent Peer Review file · Nature Communications]

Facilitating Alkaline Hydrogen Evolution Kinetics via Interfacial Modulation of Hydrogen-Bond Networks by Porous Amine Cages

Corresponding Author: Professor Jiayin Yuan

Version 0:

Reviewer comments:

Reviewer #1

(Remarks to the Author)

In this manuscript, the authors reported a porous amine cage-confined Pt cluster as a target-specific model to engineer the electrochemical interface of alkaline hydrogen evolution reaction. This Pt/cage catalyst shows improved performance for alkaline HER. By HER kinetics analysis, the origin of the enhancement was attributed to the eased hydrogen-bond network by the amine groups, which promote the proton transfer in the interfacial layers. The authors conducted in situ electrochemical spectroscopic experiments to probe the interfacial water structure, which proved the weakened hydrogen-bond network experimentally. With the assistance of AIMD simulation, the weakened hydrogen-bond network by cage interruption was confirmed theoretically. Moreover, the amine groups were reported as the pump for proton transfer by forming and breaking hydrogen-bonds via the eased hydrogen-bond network, which largely lowered the kinetic barrier for alkaline HER in this work. This work is well presented, and the strategy reported in this work provide a new insight into the engineering of the electrochemical interfaces which is of high interest for the electrocatalysis field. Thus, I would recommend the publication of this manuscript in Nature Communications after minor revision regarding the following concerns:

1. 'The Pt existed as ultrafine clusters in an average size of ~0.50 nm in a narrow size distribution, highlighting the physical confinement effect by the cage frame, in drastic contrast to the control Pt sample synthesized similarly except without adding the cage'. The configuration of the Pt cluster by porous amine cage was revealed by comparing the size distribution of Pt/cage and Pt without cage. Is there any experiment that could prove Pt cluster was confined in the cage? The author may need provide more experimental data to support this.
2. How about the alkaline HER performance of Pt that prepared without adding cage? The author may need also compare its performance.
3. Does the Pt cluster-confined in cage changes after performance measurements? More post characterizations, such as XRD and TEM analysis, need to be provided.
4. In situ Raman were conducted to probe the interfacial water structure. 'Laser with an excitation wavelength of 785 nm'. The laser with the wavelength of 532 nm or 637.8 nm are usually selected. Why the author use 785 nm wavelength? Did the author try laser with wavelength of 532 nm to get better resolution of the Raman spectra?
5. The electrochemical interfaces of HER were under intensive investigation with crystalline Pt as the electrodes, mostly based on Pt(111) by previous reported literatures. The novelty of this Pt cluster structure should be further clarified. Also, Pt(111) is the dominated facet in polycrystalline Pt. Why the author choose Pt(100) as the comparing model for simulation in this work? Is there any specific reason regarding this?

Reviewer #2

(Remarks to the Author)

In this manuscript (NCOMMS-24-29917), Zhou et al. investigated interfacial modulation of the hydrogen-bond network to facilitate alkaline hydrogen evolution kinetics by using a porous amine cage as an interfacial modifier to Pt clusters in a confining configuration. They employed in situ SERS and AIMD simulation to elucidate that the interaction between water and the -NH- moiety of the cage frame largely softens the hydrogen bonds in interfacial water during HER, resulting in flexible reorganization of the interfacial water network to facilitate charge transfer. Although these results provide a molecular-scale understanding of engineering electrochemical interfaces to promote reaction kinetics, many similar studies have been reported in recent years, especially regarding water restructuring in HER, thereby significantly diminishing the

novelty of this work. Moreover, some crucial data were not processed correctly, leading to arbitrary conclusions. Therefore, I cannot recommend the publication of this work.

1. The authors did not provide the correct Pt L3-edge XANES spectra corresponding to Pt/cage, Pt foil, and PtO₂ in Fig. 1d. Additionally, based on the white-line intensity, the intensity of the Pt/cage sample was between those of Pt foil and PtO₂ reference, indicating a higher oxidation state of Pt in the Pt/cage catalyst, instead of the Pt⁰ claimed by the authors in lines 141-142.

2. The reviewer also found that the fitting for the Pt 4f XPS spectra in Fig. S5 was incorrect. Please explain why the peak for Pt⁰ in Pt/cage exhibited such a significant shift toward higher binding energies compared to previously reported results. Moreover, the existing species/oxidation states based on XPS results were inconsistent with XAS data. Additionally, the authors did not correctly assign the peak at 74 eV in the Pt/C sample.

3. The authors should recognize that a well-designed control sample is necessary to avoid arbitrary conclusions. For instance, based on the similar shape and edge energy of Pt/cage and Pt foil, they claimed that "little-to-no electronic structure change" occurred in the Pt cluster due to the cage confinement. However, comparing XPS spectra between Pt/cage and Pt/C samples was inappropriate due to the large size difference. Similarly, the control samples for electrochemical testing were also not appropriate (lines 187-191). A more reasonable approach might be preparing a physical-mixing sample by adding an equivalent amount of pure cage into the Pt cluster (instead of Pt/C in this work) catalyst ink.

4. The EXAFS fitting in Fig. 1e was not provided in a rigorous manner. Shoulder peaks were evident at 2.8-4 Å, yet the authors ignored them and claimed exclusive Pt-Pt coordination in Pt clusters of Pt/cage. The clear difference between Pt/cage and Pt foil can also be distinguished in WT-EXAFS spectra (Fig. 1f). It is reasonable to assume that some oxidized species are present at uncoordinated sites in ultrafine Pt clusters.

5. The main concern of this work is the stability of Pt clusters during HER, which significantly affects reaction kinetics. It is well known that uncoordinated Pt sites on the cluster surface remain reactive and unstable at highly cathodic potentials. However, corresponding experimental evidence and discussions were missing. In situ characterizations are necessary for assessing structural stability. The authors claimed no change in the valence state of Pt after cycles, but this result does not validate structural robustness.

6. To investigate the interactions between the cage and interfacial water during the alkaline HER process, in situ SERS analysis of the physical-mixing control sample prepared by adding an equivalent amount of pure cage into the Pt cluster catalyst should be provided.

7. For the in situ SERS spectra of Pt/cage (Fig. 3c), the authors should clearly elucidate the inversed peak shift of strong H-bonded water, variations in the shifts of other peaks, and the significantly decreased intensity at -0.3 V.

8. Considering the uncertain structure of Pt/cage in this work (as discussed above), the model used for AIMD simulation and the mechanistic understandings are not convincing.

9. The authors should thoroughly review previous studies on this research topic and clearly state the differences and contributions of this work in the Introduction.

10. There are numerous typographical errors in the manuscript. For instance, on line 130, page 6, the TEM image of the control Pt cluster sample (synthesized without adding the cage) was displayed in Supplementary Fig. 3 instead of Fig. 3. The authors should carefully check for such errors throughout the manuscript.

Reviewer #3

(Remarks to the Author)

The paper "Interfacial modulation of hydrogen-bond network by porous amine cage for facilitating alkaline hydrogen evolution kinetics" reports the synthesis, characterization, AIMD simulations and HER kinetics of Pt/cage. The authors presented the experimental as well as computational work in detail. The manuscript is well organized and well written. The work presented is of good standard and can attract the scientific community of the journal. I'd like to recommend this manuscript to be published in Nature Communications after considering following minor points:

1. Authors should clearly mention the heading of each section to differentiate everything. Please make changes as per the guidelines of the journal.

2. The structural geometry of the theoretical study needs to be included. Discuss the theoretical results and compare them with experimental details.

3. The visibility of all figures should be improved by using larger line sizes and/or bold fonts.

4. An in-depth comparison of the results with previous similar work is missing. The authors should prepare a comparison table that includes all key findings and values compared with similar works.

5. The results value of overpotential from DFT simulation needs to be discussed.

6. Some typographical errors are found in the manuscript and should be corrected.

7. In Fig. 4e and 4g, the author needs to run AIMD simulation for 7ps -9ps for better insight into bond length evolution. There are many references which discuss about AIMD. For AIMD authors can see the following papers
doi.org/10.1016/j.ijhydene.2022.08.084

8. Although, the author calculated kinetic barrier for Volmer step, the author recommended to run climbing-image nudged elastic band calculation for complete hydrogen evolution reaction to gain detailed insight into energetics of the reaction and to obtain exact reaction barrier to analyze accurate catalytic activity.

Version 1:

Reviewer comments:

Reviewer #1

(Remarks to the Author)

The authors have addressed my concerns. Please accept as is.

Reviewer #2

(Remarks to the Author)

For comment 1, authors did not correctly understand how to reasonably evaluate the oxidation state from XANES spectra. Thus, they should pay more attention to it. Firstly, they claimed that “the white line intensity—when it varies consistently with the energy edge shift—can provide indirectly insights into the valence state of the absorbing atom in samples. However, when the changes in edge energy and white-line intensity are not consistent, one needs to be cautious about using white-line intensity alone to infer changes in valence state, especially when the target sample is measured in a different mode from the reference sample.”. These mentions are not correct. For Pt case, the dipole-allowed transitions to vacant localized d states result in an intense feature (so-called white line), thus the intensity of white line, instead of absorption edge position, is in fact a direct indicator for evaluate the oxidation state (d state) of Pt. Note particularly that, the shift of edge position might be not reliable for predicting oxidation state because it is strongly dependent on the atomic configuration/structure of target element.

Secondly, authors explained that the XAS measurement of Pt/cage sample in this work might be significantly influenced by the self-absorption using a total-yield fluorescence detector. Notably, despite it is often overlooked in current studies, self-absorption during XAS measurement in fluorescence mode should be carefully avoided. Once self-absorption occurs, XAS spectra might provide misleading information.

Thus, the oxidation state extracted from current XAS spectra is not convincing in this work. The authors are suggested to provide correct spectra and understanding.

Reviewer #3

(Remarks to the Author)

Manuscript is acceptable for publication in its current form.

Version 2:

Reviewer comments:

Reviewer #2

(Remarks to the Author)

The authors have addressed the questions raised by the reviewer. The manuscript can be accepted in its current form.

Response letter to Editor and Reviewers

Comments from Reviewer #1 (Remarks to the Author):

General comments: *In this manuscript, the authors reported a porous amine cage-confined Pt cluster as a target-specific model to engineer the electrochemical interface of alkaline hydrogen evolution reaction. This Pt/cage catalyst shows improved performance for alkaline HER. By HER kinetics analysis, the origin of the enhancement was attributed to the eased hydrogen-bond network by the amine groups, which promote the proton transfer in the interfacial layers. The authors conducted in situ electrochemical spectroscopic experiments to probe the interfacial water structure, which proved the weakened hydrogen-bond net experimentally. With the assistance of AIMD simulation, the weakened hydrogen-bond network by cage interruption was confirmed theoretically. Moreover, the amine groups were reported as the pump for proton transfer by forming and breaking hydrogen-bonds via the eased hydrogen-bond network, which largely lowered the kinetic barrier for alkaline HER in this work. This work is well presented, and the strategy reported in this work provide a new insight into the engineering of the electrochemical interfaces which is of high interest for the electrocatalysis field. Thus, I would recommend the publication of this manuscript in Nature Communications after minor revision regarding the following concerns:*

Response: Thank you very much for your positive feedback. We have carefully addressed each of your comments and made the corresponding revisions in the updated manuscript.

Comment 1. *'The Pt existed as ultrafine clusters in an average size of ~0.50 nm in a narrow size distribution, highlighting the physical confinement effect by the cage frame, in drastic contrast to the control Pt sample synthesized similarly except without adding the cage'. The configuration of the Pt cluster by porous amine cage was revealed by*

comparing the size distribution of Pt/cage and Pt without cage. Is there any experiment that could prove Pt cluster was confined in the cage? The author may need provide more experimental data to support this.

Response: Thank you for raising this important point. Characterization of the tiny-sized metal cluster (Pt, Au, or Pd) inside a cage is a common question and has been addressed to the same cationic cage (C-Cage) in our previous studies (Chem. Sci. 2019, 10, 1450; Cell Rep. Phys. Sci. 2021, 2, 100546; and Nature Commun., 2022, 13, 1471.). These metal cluster-encapsulated cationic cages were all prepared in the same way by reduction of the metal complex anions that are associated with the cationic cage in aqueous solution. A concrete proof of the physical confinement of the metal cluster inside the cationic cage (termed C-Cage) was given in our paper Chem. Sci. 2019, 10, 1450, where the Au cluster encapsulated C-Cage (termed as Au@C-Cage) was used as a model example. We first used HAADF-STEM to determinate the size of Au clusters (0.65 ± 0.2 nm) that matched well with the pore size of C-Cages (~ 0.72 nm). In addition, the size of Au@C-Cage determined by cryo-EM data and DLS was consistent with single native C-Cage, thus excluding the possibility of a single Au cluster stabilized by two or multiple cages. Then, we used the NMR technique to directly analyze Au@C-Cage and specifically the spatial relationship between the cage host and the Au cluster. Here we observed the difference of proton signals in ^1H -NMR spectra between pristine C-Cage and Au@C-Cage, i.e., the observed broadening of all peak widths in the ^1H -NMR spectrum of Au@C-Cage in comparison to pristine C-Cage. This difference supports a local restricted motion and structural heterogeneity stemming from encapsulation of the Au cluster. Then, 2D DOSY ^1H -NMR further showed similar diffusion coefficients for Au@C-Cage ($2.16 \times 10^{-6} \text{ cm}^2 \text{ s}^{-1}$) and pristine C-Cage ($2.06 \times 10^{-6} \text{ cm}^2 \text{ s}^{-1}$), and confirmed the similar size and shape of native C-Cage and Au@C-Cage. These observations proved that metal cluster rested inside the cage cavity instead of intercage interactions or aggregation on the cage surface. We are not the sole group drawing this conclusion, as similar observations were reported in other metal cluster@cage systems (J. Am. Chem. Soc. 2014, 136, 1782; Nat. Catal. 2018, 1, 214; J.

Am. Chem. Soc. 2018, 140, 13835.).

Accordingly, in the revised manuscript, we have added texts to clarify this point and referred the approach to prove the encapsulation of the metal cluster inside the C-Cage to our previous studies. “The successful encapsulation of the metal cluster into cage has been well demonstrated in our previous work.³⁵⁻³⁷”

Comment 2. *How about the alkaline HER performance of Pt that prepared without adding cage? The author may need also compare its performance.*

Response: Thank you for your suggestion. The synthesized Pt without the addition of cage (referred to as ‘bare Pt’) formed Pt nanoparticles with an average size of 2.5 ± 0.3 nm and exhibited obvious agglomeration due to the lack of confinement provided by the cage (Supplementary Figure 3). The electrochemical hydrogen evolution performance of bare Pt was evaluated with the same Pt loading as Pt/C. As shown in **Figure R1**, the bare Pt demonstrated lower HER activity, with an overpotential of 69 mV at 10 mA cm^{-2} , comparable to the commercial Pt/C, which had an overpotential of 64 mV at 10 mA cm^{-2} . This reduced activity is probably due to the agglomeration of Pt nanoparticles, which limits mass transport and reduces the availability of exposed Pt surface sites.

Figure R1. HER performance of Pt/C, Pt with and without adding cage.

Table R1. The weight contents of Pt by ICP-OES and the electrode mass loadings of Pt/cage, Pt/C, and bare Pt.

Sample	Weight content	Mass loading ($\mu\text{g}_{\text{Pt}}/\text{cm}_{\text{geo}}^{-2}$)
Pt/cage	5.30 wt. %	5.4
Pt/C	20.0 wt. %	5.1
Bare Pt	98.2%	5.1

Comment 3. Does the Pt cluster-confined in cage changes after performance measurements? More post characterizations, such as XRD and TEM analysis, need to be provided.

Response: We appreciate the reviewer's constructive suggestion. We performed the post-characterization analyses following the stability tests to examine the robustness of the Pt/cage structure. As shown in **Figure R2**, no obvious crystalline Pt was observed after the stability test.

Figure R2. X-ray diffraction patterns of Pt/cage before and after stability test.

Additionally, the HAADF-STEM imaging (**Figure R3**) revealed that the size of the Pt clusters in the tested Pt/cage sample remained practically unchanged, indicating the structural stability of the cage-confined Pt clusters. These findings are consistent with the X-ray photoelectron spectroscopy result presented in Supplementary Figure 2. The post-characterization data and corresponding analysis have been added to the supporting information and highlighted in the revised manuscript.

Figure R3. HAADF-STEM image of Pt/cage after stability test.

Comment 4. *In situ Raman were conducted to probe the interfacial water structure. ‘Laser with an excitation wavelength of 785 nm’. The laser with the wavelength of 532 nm or 637.8 nm are usually selected. Why the author use 785 nm wavelength? Did the author try laser with wavelength of 532 nm to get better resolution of the Raman spectra?*

Response: Yes, we have tried the *in situ* electrochemical surface-enhanced Raman measurements using a 532 nm laser. However, fluorescence effects were observed with this high-frequency laser excitation, which interfered with the spectral analysis. After switching to the lower-frequency laser with a 785 nm wavelength, we achieved good resolution in the measured spectra without fluorescence interference. Therefore, we used the 785 nm laser for irradiation in our experiments.

Comment 5. *The electrochemical interfaces of HER were under intensive investigation with crystalline Pt as the electrodes, mostly based on Pt(111) by previous reported literatures. The novelty of this Pt cluster structure should be further clarified. Also, Pt(111) is the dominated facet in polycrystalline Pt. Why the author choose Pt(100) as the comparing model for simulation in this work? Is there any specific reason regarding this?*

Response: We thank the reviewer for this good question. The Pt(111) single-crystal electrode is widely used as a classic model for investigating pH-dependent HER kinetics at electrochemical interfaces. Recent studies have shown that the interfacial water structure plays a key role in the reorganization of water net, which impacts charge

transfer process and, therefore the HER kinetics. Many studies are now focusing on enhancing the flexibility and connectivity of the interfacial water network. In these studies, Pt(111) single-crystal electrodes are often modified with surface promoters like Ni(OH)₂ or Ru to steer interfacial water structure. However, these surface promoters also alter or partially occupy the surface sites of Pt.

An alternative approach involved introducing organic additives into the electrolyte to directly interact with the interfacial water. Unfortunately, these organic additives often introduce steric hindrance, which impedes mass transfer on the Pt surface. Drawing inspiration from these studies and recognizing the challenges, we developed an organic porous modifier to modify the Pt surface in a confining configuration. This configuration fully exposes the Pt surface sites while minimizing steric effect due to the open windows of the 3D porous cage. Additionally, the proximity of the cage modifier to the Pt surface ensures precise interactions with the interfacial water. Furthermore, the confinement effect by cages also induces the formation of ultrafine Pt clusters, which increase the number of exposed Pt sites, thereby enhancing HER activity and improving atomic utilization of the precious Pt metal. In summary, the cage-confined Pt cluster structure not only enables precise modulation of the Pt-electrolyte interface without occupying Pt surface sites or causing steric hindrance, but also maximizes the number of exposed Pt sites for HER.

We agree with the reviewer that Pt(111) is the dominant facet in polycrystalline Pt. However, when considering electrochemical reactions in an electrolyte, Pt(111) represents a specific case rather than a representative one. First, Pt(111) does not exhibit the characteristic H_{UPD} peak in cyclic voltammetry curve, which is commonly observed on other facets (*Nat. Energy* **2017**, *2*, 17031). Moreover, the HER kinetics of Pt(111) shows significant pH-dependence, but its H_{UPD} peak hardly shifts with pH. Furthermore, the HER activity of Pt(111) is independent on cation identity (such as Li⁺, Na⁺, and K⁺) (*ACS Meas. Sci. Au* **2021**, *1*, 48-55), likely due to the saturated coordination of Pt(111), which limits its ability to bind HER intermediates such as OH. Therefore, we have chosen the Pt(100) facet as our model to better understand the H adsorption and charge

transfer behavior on its surface.

Comments from Reviewer #2 (Remarks to the Author):

General comments: In this manuscript (NCOMMS-24-29917), Zhou et al. investigated interfacial modulation of the hydrogen-bond network to facilitate alkaline hydrogen evolution kinetics by using a porous amine cage as an interfacial modifier to Pt clusters in a confining configuration. They employed in situ SERS and AIMD simulation to elucidate that the interaction between water and the -NH- moiety of the cage frame largely softens the hydrogen bonds in interfacial water during HER, resulting in flexible reorganization of the interfacial water network to facilitate charge transfer. Although these results provide a molecular-scale understanding of engineering electrochemical interfaces to promote reaction kinetics, many similar studies have been reported in recent years, especially regarding water restructuring in HER, thereby significantly diminishing the novelty of this work. Moreover, some crucial data were not processed correctly, leading to arbitrary conclusions. Therefore, I cannot recommend the publication of this work.

Response: We thank the reviewer for the constructive comments, which have helped us improve our manuscript. We have responded to each comment point by point, and the necessary revisions have been incorporated into the updated manuscript and supporting information. We hope our endeavor in improving the manuscript will upgrade our work to the standard expected by the Reviewer for this journal.

Comment 1. The authors did not provide the correct Pt L3-edge XANES spectra corresponding to Pt/cage, Pt foil, and PtO₂ in Fig. 1d. Additionally, based on the white-line intensity, the intensity of the Pt/cage sample was between those of Pt foil and PtO₂ reference, indicating a higher oxidation state of Pt in the Pt/cage catalyst, instead of the Pt⁰ claimed by the authors in lines 141-142.

Response: Thank you for careful reading of the manuscript. We apologize for the inappropriate labelling of the sample name in Fig. 1d. We have corrected the sample names, which are now consistent with that in Fig. 1e.

Figure R4. XANES spectra at Pt L3-edge for Pt/cage, Pt foil, and PtO₂.

As for the white-line intensity, we agree with the reviewer that, in addition to the energy shift of the edge, the white line intensity—when it varies consistently with the energy edge shift—can provide indirectly insights into the valence state of the absorbing atom in samples. However, when the changes in edge energy and white-line intensity are not consistent, one needs to be cautious about using white-line intensity alone to infer changes in valence state, especially when the target sample is measured in a different mode from the reference sample. In our case, the Pt foil sample and PtO₂ samples were measured in transmission mode, while Pt/cage sample was measured in fluorescence mode (with a reference sample) using a total-yield fluorescence detector. A known issue with fluorescence mode is the self-absorption, where the absorption intensity is affected by scattering and fluorescence from elements other than the absorbing element (Jeroen A. van Bokhoven et al., *X-Ray Absorption and X-Ray Emission Spectroscopy: Theory and Applications*, **2016**; Scott Calvin et al., *XAFS for everyone*, **2013**, <https://doi.org/10.1201/b14843>; *Current Opinion in Electrochemistry* **2021**, *30*, 100803; *Phys. Sci. Reviews* **2020**, 20170181; *J. Synchrotron Rad.* **2005**, *12*, 537-541). Self-absorption correction is helpful to some extent but is only approximate, as it depends not only on fluorescence mode but also on the thickness and concentration of the sample. Therefore, here it is more reliable and safer to extract valence state information from the position of the white line rather than its intensity (though the intensity can be indeed informative in some other cases). The self-absorption effect should be taken into consideration when processing and interpreting XAS data, yet

often overlooked.

Moreover, it has been reported that white-line intensity varies significantly for small cluster sizes and specific geometry, but becomes size-independent for large cluster (*J. Chem. Phys.* **2002**, *116*, 1911-1919). Given the ultrafine size of the Pt clusters in Pt/cage sample in this work, we rely on the edge positions (either halfway up the edge or the energy at the top of the white line) to cautiously compare the valence states. This approach has also been used in a similar case in previously published literature (*Nat. Commun.* **2024**, *15*, 5998).

Comment 2. The reviewer also found that the fitting for the Pt 4f XPS spectra in Fig. S5 was incorrect. Please explain why the peak for Pt⁰ in Pt/cage exhibited such a significant shift toward higher binding energies compared to previously reported results. Moreover, the existing species/oxidation states based on XPS results were inconsistent with XAS data. Additionally, the authors did not correctly assign the peak at 74 eV in the Pt/C sample.

Response: We appreciate the reviewer's kind reminder. The XPS data, calibrated from the C 1s signal, have been updated (**Figure R5**). The dominant XPS peak in the commercial Pt/C sample corresponds to Pt⁰ (*Nat. Commun.* **2023**, *14*, 1711; *Nat. Commun.* **2019**, *10*, 3808), and the dominant XPS peak in the Pt/cage sample aligns well with that in the commercial Pt/C sample. There is no oxidative specie observed in the XPS results of Pt/cage. The slight shift observed is likely due to the high surface energy of the Pt clusters, making them more oxophilic when exposed to air. The XPS result further validates our XAS findings, confirming that the Pt in Pt/cage exists primarily as Pt⁰.

Figure R5. X-ray photoelectron spectra of Pt/cage and commercial Pt/C after C 1s calibration.

Comment 3. The authors should recognize that a well-designed control sample is necessary to avoid arbitrary conclusions. For instance, based on the similar shape and edge energy of Pt/cage and Pt foil, they claimed that "little-to-no electronic structure change" occurred in the Pt cluster due to the cage confinement. However, comparing XPS spectra between Pt/cage and Pt/C samples was inappropriate due to the large size difference.

Similarly, the control samples for electrochemical testing were also not appropriate (lines 187-191). A more reasonable approach might be preparing a physical-mixing sample by adding an equivalent amount of pure cage into the Pt cluster (instead of Pt/C in this work) catalyst ink.

Response: We thank the reviewer for the comment. As for the XAS characterization, we used Pt foil as the reference sample for the calibration of edge energy. Employing a reference metal foil of the same element to minimize the deviation from the monochromator is the default practice in XAS measurement. As for XPS measurements, we used the commercial Pt/C sample as its valence state (Pt^0) is well established. In this context, we see no conflict in the selection of control samples between these two measurements.

Regarding the control sample for electrochemical tests, we agree that bare Pt would be ideal for comparison. However, synthesizing bare Pt below 1 nm without a protective layer is in our view hard to realize. Protection layer, such as surface ligands, would occupy the active surface sites of the Pt clusters, leading to more ambiguous comparison, which we aim to avoid. We synthesized the Pt clusters using the same procedure as for Pt/cage, but without the cage. HAADF-STEM image shows that the size of these "bare Pt" is averagely around 3 nm (**Figure R6**), similar to that of commercial Pt/C.

Figure R6. HAADF-STEM image of bare Pt nanoparticles synthesized without adding cage (2.5 ± 0.3).

Figure R7. HAADF-STEM image of Pt/cage supported on graphite nanoplatelets after removing cage layer by pyrolysis (3.1 ± 0.3 nm).

We did try to produce Pt clusters without cages by other methods. For example, we attempted to immobilize Pt clusters onto supporting materials. The as-prepared Pt/cage sample was sonicated with graphite nanoplatelets and then pyrolyzed in air (with a low heating rate to 200°C) to remove the cage layer. However, the Pt clusters agglomerated upon removal of the cage during the pyrolysis process (**Figure R7**). Consequently, we chose commercial Pt/C as the control sample. Additionally, we compared the electrocatalytic HER performance of Pt/C with and without the cage, which demonstrated that the confining configuration is important to allow the cage to interact with interfacial water (Supplementary Fig. 10).

Comment 4. The EXAFS fitting in Fig. 1e was not provided in a rigorous manner.

Shoulder peaks were evident at 2.8-4 Å, yet the authors ignored them and claimed exclusive Pt-Pt coordination in Pt clusters of Pt/cage. The clear difference between Pt/cage and Pt foil can also be distinguished in WT-EXAFS spectra (Fig. 1f). It is reasonable to assume that some oxidized species are present at uncoordinated sites in ultrafine Pt clusters.

Response: We thank the reviewer for pointing out these concerns. The Fourier transformed EXAFS spectrum in Fig. 1e aims to investigate the existence of Pt-Pt and Pt-O paths in the Pt/cage sample by comparing it with reference samples of Pt foil and PtO₂. The spectrum of PtO₂ clearly shows Pt-O coordination at a R value around 1.8, which we do not observe in the Pt/cage sample. It is important to note that signals for Pt-C, N, or O coordination appear at shorter R values than Pt-Pt (*Nat. Catal.* **2019**, *2*, 495-503; *Nat. Commun.* **2017**, *8*, 15938), which further supports the absence of these interactions in the case of Pt/cage sample.

Furthermore, the Pt cluster in Pt/cage sample, with such an ultrafine size, is not crystalline. Even if they were crystalline, the Pt-Pt bond lengths can vary across different facets, suggesting there must be differences in Pt-Pt bond lengths. Considering these factors, the shoulder peak near the dominant signal, in our opinion, likely corresponds to a Pt-Pt shell with slightly longer bond lengths, as well as a Pt-Pt-Pt shell. Numerous studies have reported variations in Pt-Pt bond lengths, supporting our observations (*Nanoscale* **2014**, *6*, 1153-1165).

Comment 5. The main concern of this work is the stability of Pt clusters during HER, which significantly affects reaction kinetics. It is well known that uncoordinated Pt sites on the cluster surface remain reactive and unstable at highly cathodic potentials. However, corresponding experimental evidence and discussions were missing. In situ characterizations are necessary for assessing structural stability. The authors claimed no change in the valence state of Pt after cycles, but this result does not validate structural robustness.

Response: Thank you for your suggestion. Given the ultrafine size of the Pt clusters, HAADF-STEM is essential for visualization. However, even with HAADF-STEM, obtaining a clear image of the Pt cluster is challenging, as the organic cage tends to decompose under the high-energy electron beam, which can blur the image. Therefore, *in situ* observation using TEM is hard to reach in our case. To assess structural robustness, we examined the size of Pt clusters after the stability test using HAADF-STEM (**Figure R3**). Within the experimental error, we see no change in size in the HAADF-STEM post-characterization, which underscored the confining and stabilizing benefits of the cage.

Figure R3. HAADF-STEM image of Pt/cage after stability test.

Comment 6. To investigate the interactions between the cage and interfacial water during the alkaline HER process, *in situ* SERS analysis of the physical-mixing control sample prepared by adding an equivalent amount of pure cage into the Pt cluster catalyst should be provided.

Response: Thank you for your comment. We have provided a detailed explanation for selecting Pt/C as the control sample for evaluating electrochemical HER performance under “*Comment 3*”. Consequently, Pt/C was chosen as the control sample for the *in situ* electrochemical SERS measurements as well.

Comment 7. For the *in situ* SERS spectra of Pt/cage (Fig. 3c), the authors should clearly elucidate the inversed peak shift of strong H-bonded water, variations in the shifts of other peaks, and the significantly decreased intensity at -0.3 V.

Response: Due to the manipulation of the cage with a reduced dipole, the Stark effect in the Pt/cage sample is suppressed, while it is more pronounced in the Pt/C sample. It is important to note that during the *in situ* electrochemical SERS measurements for HER, the population of interfacial water probed at a fixed confocal laser depth can be influenced by the generated H₂ bubbles, particularly at more negative potentials. Therefore, the three types H-bonded water were compared by their ratios rather than their absolute intensities or absolute populations.

Comment 8. Considering the uncertain structure of Pt/cage in this work (as discussed above), the model used for AIMD simulation and the mechanistic understandings are not convincing.

Response: We have addressed the above comments in details, point by point. Please refer to our above responses for these comments. With the changes and clarification made in this revision round, we hope to convince Reviewer about the structure of Pt/cage. The AIMD model used in this work was selected carefully based on our characterization results.

Comment 9. The authors should thoroughly review previous studies on this research topic and clearly state the differences and contributions of this work in the Introduction.

Response: Thank you for your suggestions. The introduction has been refined to better highlight the contributions of this work. Please refer to the highlighted part in the revised manuscript.

Comment 10. There are numerous typographical errors in the manuscript. For instance, on line 130, page 6, the TEM image of the control Pt cluster sample (synthesized without adding the cage) was displayed in Supplementary Fig. 3 instead of Fig. 3. The authors should carefully check for such errors throughout the manuscript.

Response: We thank the reviewer for careful reading. We have thoroughly and carefully reviewed and revised the entire manuscript.

Comments from Reviewer #3 (Remarks to the Author):

General comments: The paper “Interfacial modulation of hydrogen-bond network by porous amine cage for facilitating alkaline hydrogen evolution kinetics” reports the synthesis, characterization, AIMD simulations and HER kinetics of Pt/cage. The authors presented the experimental as well as computational work in detail. The manuscript is well organized and well written. The work presented is of good standard and can attract the scientific community of the journal. I’d like to recommend this manuscript to be published in Nature Communications after considering following minor points:

Response: We appreciate the reviewer’s recognition of our work here and positive comments. All of the suggestions have been addressed point by point, and the necessary changes have been made accordingly in the revised manuscript.

Comment 1. Authors should clearly mention the heading of each section to differentiate everything. Please make changes as per the guidelines of the journal.

Response: Thank you for your kind suggestion. We have added the headings in a highlighting form to each section in accordance with the journal guidelines.

Comment 2. The structural geometry of the theoretical study needs to be included. Discuss the theoretical results and compare them with experimental details.

Response: Thank you for your constructive suggestion. We have uploaded the file containing the structural geometry of the AIMD models. Additionally, we have expanded the discussion on theoretical and experimental results in the AIMD section of the revised manuscript, highlighted by the statements: “Next, the probability distribution function of H-bonds at different lengths with and without cage was compared to get insight into the interaction...”, and “The comparison of the calculated kinetic barriers for individual hydrogen adsorption reactions further verified the promoted kinetics of rate-determining Volmer step...”.

Comment 3. The visibility of all figures should be improved by using larger line sizes and/or bold fonts.

Response: Thank you for your thoughtful suggestion. All figures have been updated for the better clarity and readability.

Comment 4. An in-depth comparison of the results with previous similar work is missing. The authors should prepare a comparison table that includes all key findings and values compared with similar works.

Response: Thank you for your constructive suggestion. We have compared the key values with previous works in Supplementary Table 5, and discussed the key finding in the introduction part with highlight “The Pt(111) single-crystal electrode is a classic model for investigating pH-dependent HER kinetics at electrochemical interfaces”, and “Specifically, Pt(111) single-crystal electrodes are often modified with surface promoters like Ni(OH)₂ or Ru to steer interfacial water structure...”.

Comment 5. The results value of overpotential from DFT simulation needs to be discussed.

Response: We thank the reviewer for this comment. The theoretical overpotentials were analyzed based on the methods reported before (*J. Phys. Chem. C* **2010**, *114*, 18182-18197; *ChemCatChem* **2011**, *3*, 1159-1165; *J. Phys. Chem. C* **2011**, *115*, 19311-19319; *ACS Catal.* **2022**, *12*, 8404-8433):

$$G^{HER} = \max(\Delta G_1, \Delta G_2)$$
$$\eta^{HER} = G^{HER}/e - 0.765V$$

Where ΔG_1 and ΔG_2 are the differences of the energies of reaction (1) and (2), respectively. an ideal catalyst should be able to facilitate the HER just above the equilibrium potential, but it requires all the two electron-transfer steps to have reaction free energies of the same magnitude at zero potential (i.e., $1.53 \text{ eV}/2 = 0.765 \text{ eV}$). This is equivalent to all the reaction free energies being zero at the equilibrium potential, 0.765V. the reaction energy of the reaction (0) in the alkaline environment can be

obtained as 1.53 eV.

Figure R8. Standard free energy diagram for HER on (a) Pt(100) and (b) Pt/cage at different electrode potential U .

As the **Figure R8 a** shows, for hydrogen evolution reaction on Pt(100), the HER sub steps are all uphill when the electrode potential U is 0 V, which corresponds to a short-circuit state. At $U = 0.765 \text{ V}$, the Volmer step remains uphill while the other elementary step becomes downhill. Only when the electrode potential increases to 1.06 V can all the elementary steps become downhill. Therefore, the information is obtained that the Volmer step is the rate-determining step and the theoretical overpotential is $1.060 \text{ V} - 0.765 \text{ V} = 0.295 \text{ V}$.

On Pt/cage (**Figure R8 b**), at applied electrode potential $U = 0.765 \text{ V}$, the Volmer step becomes downhill while Heyrovsky step remains uphill. All the elementary steps become downhill until the electrode potential increases to 0.890 V. This reveals that the first electron-transfer Volmer step is facilitated and Heyrovsky step becomes the rate-determining step. The overpotential is lowered to $0.890 \text{ V} - 0.765 \text{ V} = 0.125 \text{ V}$ on Pt/cage, which is much lower than that on Pt(100) and consistent with our experimental results.

Comment 6. Some typographical errors are found in the manuscript and should be corrected.

Response: Thank you for your kind reminder. We sincerely apologize for our

carelessness. The manuscript and supporting information have been carefully checked and revised thoroughly.

Comment 7. In Fig. 4e and 4g, the author needs to run AIMD simulation for 7ps -9ps for better insight into bond length evolution. There are many references which discusses about AIMD. For AIMD authors can see the following papers
doi.org/10.1016/j.ijhydene.2022.08.084

Response: We thank the reviewer for the professional suggestion and fully agree that a longer running time might provide better insight into bond length evolution for AIMD simulations. However, the optimal running time can vary based on the specific case and practical considerations. In our study, as shown in Figure 4e and g, bond breaking and forming occur between 450 fs to 750 fs, completing the evolution within a 300 fs window both on Pt(100) and Pt/cage models. Moreover, the bond lengths stabilize beyond this 300 fs window throughout the simulation. We set the total running time to 1200 fs, as no further changes corresponding to bond breaking or forming were observed after 750 fs.

To carefully justify our choice of the running time, we reviewed relevant literature. We found that Professor William A. Goddard III and colleagues used a similar simulation approach in their recent work (*Nat. Energy* **2023**, 8, 859-869), applying a running time of 1000 fs to analyze the bond evolution, which was deemed sufficient to support their conclusions. Other studies also reported comparable running times that led to convincing findings in AIMD simulations (*J. Chem. Phys.* **2024**, 160, 071102).

With these considerations, we believe that our simulation with a running time of 1200 fs is reasonable and robust enough to support our conclusions. If the reviewer might believe that additional simulation time is necessary, we can conduct further simulations, but it would require a significant investment of supercomputing resources (2+ months), as the Pt/cage system is quite large for AIMD methods. The corresponding references (*Int. J. Hydrogen Energy* **2022**, 48, 37860-37871; *Nat. Energy* **2023**, 8, 859-869; *J. Chem. Phys.* **2024**, 160, 071102) due to their relevance to our AIMD simulation have

been cited in the revised manuscript.

Comments 8. Although, the author calculated kinetic barrier for Volmer step, the author recommended to run climbing-image nudged elastic band calculation for complete hydrogen evolution reaction to gain detailed insight into energetics of the reaction and to obtain exact reaction barrier to analyze accurate catalytic activity.

Response: We thank the reviewer's suggestion. Here we analyzed the free energy changes for the whole hydrogen evolution reaction with climbing-image nudged elastic band calculation method (*J. Chem. Phys.* **2000**, *113*, 9901-9904; *J. Chem. Phys.* **2000**, *113*, 9978-9985). The solvation effect was also considered with VASP sol++ for the calculation to include the interactions derived from electrostatics, cavitation, and dispersion (*J. Chem. Phys.* **2023**, *159*, 234117).

The climbing-image nudged elastic band (CI-NEB) calculation method was applied for the more accurate finding of saddle points here. The transition states of Volmer and Heyrovsky steps were searched using CI-NEB method in this work. Three images were inserted into the initial and final configurations for finding the accurate transition states.

HER can be expressed as following equation:

The HER activities on active sites were studied in details. HER occurs over the active sites under alkaline condition in the following electron-transfer Volmer and Heyrovsky paths according to our experimental Tafel slope results:

(* indicates the adsorbing site on the catalyst surface)

Where * stands for an active site on the catalyst surface, (*l*) and (*g*) refer to liquid and gas phases, respectively, and H* represents the adsorbed hydrogen atom.

In order to obtain the rate-determining step of HER on different catalysts, we calculated

the adsorption free energy of H*, according to the following equation:

$$\Delta E^*H = E(H^*) - E(*) - 1/2E_{H_2} \quad (3)$$

Where E(H*), E(*) and E_{H2} were the ground state energies of surfaces adsorbed with H*, clean surface and H₂ molecules in the gas phase, respectively. We also considered the ZPE and entropy corrections here. These calculations transform DFT binding energies, ΔE^{DFT}, into free energies of adsorption, ΔG_{ads}, by the following equation:

$$\Delta G_{ads} = \Delta E^{DFT} + \Delta ZPE - T\Delta S \quad (4)$$

where T is the temperature and ΔS is the entropy change. For the zero point energy (ZPE), the vibrational frequencies of adsorbed species were calculated with the Pt(100) and Cage-fixed to obtain ZPE contribution in the free energy expression.

For each step, the reaction free energy ΔG is defined as the difference between free energies of the initial and final states and is given by the expression:

$$\Delta G = \Delta E + \Delta ZPE - T\Delta S + \Delta G_U + \Delta G_{pH} \quad (5)$$

where ΔE is the reaction energy of reactant and product molecules adsorbed on the surface of catalyst, obtained from DFT calculations, ΔG_U = -eU, where U is the potential at the electrode, and e is the charge transferred. ΔG_{pH} was the correction of the H⁺ free energy by the concentration dependence of the entropy:

$$\Delta G_{pH} = -k_B T \ln[H^+] \quad (6)$$

where k_B is the Boltzmann constant.

The free energy of reaction (1) and (2) can be calculated using equation (5).

The solvation effect has been included by VASP sol++, which implements an implicit solvation model that describes the effect of electrostatics, cavitation, and dispersion on the interaction between a solute and solvent. The relative dielectric constant was set to 78.4, corresponding to water at room temperature.

Figure R9. Free energy diagrams of complete hydrogen evolution reaction on Pt(100) and Pt/cage. (a) Volmer step. (b) Heyrovsky step.

(* indicates the adsorbing site on the catalyst surface)

Following the Volmer step, Heyrovsky step was chosen as the second elementary step according to the Tafel slope values in our experimental results (Figure 2f) which revealed that the hydrogen evolution reaction becomes Heyrovsky-limited on Pt/cage under alkaline conditions. As shown in **Figure R9**, an energy barrier of 1.62 eV to overcome the Volmer step on Pt(100), indicating the Volmer step is the rate-determining step of the whole hydrogen evolution reaction on Pt(100). However, this barrier is largely decreased and lower than that of Heyrovsky step (1.25 eV) on Pt/cage model, revealing the easier Volmer step followed by the rate-determining Heyrovsky step on Pt/cage. The results were in good consistence with our experimental observation that indicates an acid-like RDS and remarkably promoted hydrogen adsorption reaction with the manipulation by cage structure. The calculation methods, results, and discussion were added in the revised manuscript and supporting information.

Response letter to reviewers

Comments from Reviewer #1 (Remarks to the Author):

The authors have addressed my concerns. Please accept as is.

Response: We thank the reviewer's recognition and all the constructive suggestions provided during the revision process, which have greatly improved our manuscript.

Comments from Reviewer #2 (Remarks to the Author):

For comment 1, authors did not correctly understand how to reasonably evaluate the oxidation state from XANES spectra. Thus, they should pay more attention to it. Firstly, they claimed that “the white line intensity—when it varies consistently with the energy edge shift—can provide indirectly insights into the valence state of the absorbing atom in samples. However, when the changes in edge energy and white-line intensity are not consistent, one needs to be cautious about using white-line intensity alone to infer changes in valence state, especially when the target sample is measured in a different mode from the reference sample.”. These mentions are not correct. For Pt case, the dipole-allowed transitions to vacant localized d states result in an intense feature (so-called white line), thus the intensity of white line, instead of absorption edge position, is in fact a direct indicator for evaluate the oxidation state (d state) of Pt. Note particularly that, the shift of edge position might be not reliable for predicting oxidation state because it is strongly dependent on the atomic configuration/structure of target element. Secondly, authors explained that the XAS measurement of Pt/cage sample in this work might be significantly influenced by the self-absorption using a total-yield fluorescence detector. Notably, despite it is often overlooked in current studies, self-absorption during XAS measurement in fluorescence mode should be carefully avoided. Once self-absorption occurs, XAS spectra might provide misleading information.

Thus, the oxidation state extracted from current XAS spectra is not convincing in this work. The authors are suggested to provide correct spectra and understanding.

Response: We thank the reviewer’s kind notice. Here, we revised our description previously based on general cases to more specific cases: The absorption edge position is commonly correlated to the oxidation state of the absorbing atom, especially for the K edge. As for Pt L3 edge, which is a p-to-d transition, the peak position and peak intensity of the white line in L-edge XANES are the two features that are mostly used to extract the information of element’s oxidation states.

In our case, the peak position of the Pt/cage sample is aligned with the Pt foil, which could provide the information of the oxidation state explicitly. For the white line intensity difference, as we stated in the 1st round revision of our manuscript, the white line intensity has additional correlation to the self-absorption as the Pt/cage sample was measured in the fluorescence mode using a total-yield fluorescence detector. The intensity is affected by scattering and fluorescence from these elements other than the absorbing element, *e.g.* the light elements from the cage layer in the Pt/cage sample. This issue has been reported by classic XAS tutorials (Jeroen A. van Bokhoven, *et al.*, X-Ray Absorption and X-Ray Emission Spectroscopy: Theory and Applications, **2016**; Scott Calvin et al., XAFS for everyone, **2013**, <https://doi.org/10.1201/b14843>; *Current Opinion in Electrochemistry* **2021**, *30*, 100803; *Phys. Sci. Reviews* **2020**, 20170181; *J. Synchrotron Rad.* **2005**, *12*, 537-541). Moreover, the white line intensity is also correlated to the particle size and specific geometry (*J. Chem. Phys.* **2002**, *116*, 1911-1919). In this case, the peak position is in our view more reliable to be employed to extract the information of the oxidation state of the ultrafine Pt clusters.

We totally understand the reviewer's concern and thank the reviewer's careful check. The XAS results in this work has been confirmed by repeated measurements to exclude any uncertain factors that might affect the correctness of the conclusion. With the above considerations, the ultrasmall size and specific geometries of the Pt/cage sample, and the XPS results, we could conclude that the XAS results are solid enough to support the conclusion.

Comments from Reviewer #3 (Remarks to the Author):

Manuscript is acceptable for publication in its current form.

Response: We sincerely appreciate the reviewer's professional suggestions, which have greatly enhanced our work, particularly the calculation section.

Response letter to reviewers

Comments from Reviewer #2 (Remarks to the Author):

The authors have addressed the questions raised by the reviewer. The manuscript can be accepted in its current form.

Response: We thank the reviewer's constructive and professional suggestions, which largely help us improved our manuscript.